# Single-cell RNA sequencing unravels the transcriptional network underlying zebrafish retina regeneration

Laura Celotto[1], Fabian Rost[2], Anja Machate[1], Juliane Bläsche[2], Andreas Dahl[2], Anke Weber[1], Stefan Hans[1], Michael Brand[1]*

[1]Technische Universität Dresden, CRTD - Center for Regenerative Therapies Dresden, Center for Molecular and Cellular Bioengineering (CMCB), Fetscherstraße, Dresden, Germany; [2]Technische Universität Dresden, DRESDEN-Concept Genome Center, Center for Molecular and Cellular Bioengineering (CMCB), Fetscherstraße, Dresden, Germany

## Abstract

In the lesioned zebrafish retina, Müller glia produce multipotent retinal progenitors that generate all retinal neurons, replacing lost cell types. To study the molecular mechanisms linking Müller glia reactivity to progenitor production and neuronal differentiation, we used single-cell RNA sequencing of Müller glia, progenitors and regenerated progeny from uninjured and light-lesioned retinae. We discover an injury-induced Müller glia differentiation trajectory that leads into a cell population with a hybrid identity expressing marker genes of Müller glia and progenitors. A glial self-renewal and a neurogenic trajectory depart from the hybrid cell population. We further observe that neurogenic progenitors progressively differentiate to generate retinal ganglion cells first and bipolar cells last, similar to the events observed during retinal development. Our work provides a comprehensive description of Müller glia and progenitor transcriptional changes and fate decisions in the regenerating retina, which are key to tailor cell differentiation and replacement therapies for retinal dystrophies in humans.

*For correspondence: michael.brand@tu-dresden.de

Competing interest: The authors declare that no competing interests exist.

## eLife assessment

Müller glial cells of the zebrafish retina can differentiate into all neural cell classes following injury, providing full regenerative capabilities of the zebrafish retina. This **valuable** study presents a description of transcriptional changes of Müller glia cells in the adult and regenerating retina using single-cell RNA sequencing. The overall evidence supporting the main claims of the authors is **solid**.

## Introduction

The vertebrate retina is the neural tissue in the back of the eyes that detects and processes visual stimuli prior to transmission to the brain (*Dowling, 1987*). The retina has a layered structure, with outer, inner and ganglion cell nuclear layers alternating with the outer and inner plexiform layers (*Amini et al., 2017*; *Hoon et al., 2014*; *Stenkamp, 2015*). The nuclear layers comprise the nuclei of retinal neurons, whereas the plexiform layers contain neuronal synapses. There are six distinct classes of retinal neurons and one main type of glia: retinal ganglion, cone photoreceptor, horizontal, amacrine, rod photoreceptor and bipolar cells, as well as Müller glia. This heterogeneity arises during development in the indicated order from a common pool of multipotent retinal progenitor cells (RPCs), which progressively restrict their competence to generate early and mid-late born cell fates (*Agathocleous and Harris, 2009*; *Alexiades and Cepko, 1997*; *Amini et al., 2017*; *Cepko, 2014*; *Cepko*

*et al., 1996*; *Livesey and Cepko, 2001*; *Stenkamp, 2015*; *Stenkamp, 2007*; *Turner and Cepko, 1987*; *Weber et al., 2014*; *Young, 1985*). Retinal development, structure and function are highly conserved among vertebrates, including the teleost zebrafish (*Danio rerio*) with some variations. For instance, humans and zebrafish possess a cone-dominated retina adapted for diurnal visual tasks, whereas nocturnal animals, like mice and rats, have rod-dominated retinae adapted for a dim light lifestyle (*Fleisch et al., 2011*; *Saade et al., 2013*; *Stenkamp, 2007*). Intriguingly, and in contrast to warm-blooded vertebrates like birds and mammals, adult zebrafish can fully regenerate their retinae upon lesions affecting different neuronal cell types, depending on the injury paradigm. For instance, intense light ablates preferentially light-sensing rod and cone photoreceptors in the central and dorsal retina, whereas application of neurotoxins, like ouabain or NMDA, affects mainly retinal ganglion and amacrine cells (*Bernardos et al., 2007*; *Fimbel et al., 2007*; *Powell et al., 2016*; *Sherpa et al., 2008*; *Vihtelic et al., 2006*; *Vihtelic and Hyde, 2000*; *Weber et al., 2013*). Independently of the lesion type, Müller glia (MG), acting as resident stem cells, are the primary cellular origin of regenerated cells. Specifically, zebrafish MG react to the lesion by partial dedifferentiation that takes place between 6 and 24 hr post lesion, followed by cell cycle re-entry, including up-regulation of proliferating cell nuclear antigen (PCNA), a marker of the S-G2/M cell cycle phases (*Bernardos et al., 2007*; *Fausett et al., 2008*; *Nagashima et al., 2013*; *Raymond et al., 2006*). PCNA expression in MG is initiated at 24–31 hr post lesion (*Lahne et al., 2015*; *Nelson et al., 2013*; *Thummel et al., 2008*) and can still be detected up to 2 days post lesion (*Nagashima et al., 2013*; *Nelson et al., 2013*). PCNA upregulation is followed by asymmetric cell division of the MG that produces renewed MG and multipotent RPCs, which are detected starting from 51 hr post lesion (*Gorsuch and Hyde, 2014*; *Lahne et al., 2020a*; *Lenkowski and Raymond, 2014*; *Thummel et al., 2008*; *Wan and Goldman, 2016*). Dedifferentiation of MG to a progenitor-like state is not complete, as MG upregulate progenitor-associated genes like *pax6* or *vsx2*, but retain their glial morphology. In this context, glial markers like Glial Fibrillary Acidic Protein (*gfap*) as well as glutamine synthase (*glula*) have been reported to be either retained or downregulated, depending on the lesion paradigm, the regeneration time point, the antibody and the transgenic line used to label MG (*Fimbel et al., 2007*; *Lenkowski and Raymond, 2014*; *Nagashima et al., 2013*; *Thummel et al., 2008*). However, in all reported cases, MG-derived RPCs do not express *gfap* (*Lenkowski and Raymond, 2014*; *Thummel et al., 2008*). Asymmetric cell division of MG occurs between 31 hr and 2 days post-lesion, followed by further proliferation of MG-derived RPCs that peaks at 4–5 days post lesion, depending on the injury paradigm (*Bernardos et al., 2007*; *Fausett and Goldman, 2006*; *Fimbel et al., 2007*; *Lahne et al., 2015*; *Nagashima et al., 2013*; *Nelson et al., 2013*; *Thummel et al., 2008*). Studies on zebrafish retina regeneration have so far mainly focused on very early phases of MG dedifferentiation and RPC production (*Fausett et al., 2008*; *Nelson et al., 2013*; *Ramachandran et al., 2010*; *Ramachandran et al., 2011*; *Thummel et al., 2008*; *Thummel et al., 2010*; *Wan et al., 2012*; *Zhao et al., 2014*). Recent work showed that MG initially assume a reactive state, characterized by the expression of *hmga1a* and *yap1*, prior to giving rise to proliferative, neurogenic RPCs (*Hoang et al., 2020*). Consequently, impaired Yap1 or Hmga1a function impedes MG reprogramming and neurogenesis in the regenerating retina (*Hoang et al., 2020*). In contrast, the molecular and cellular identities of self-renewed MG and MG-derived RPCs remain elusive. RPCs have fusiform nuclei that closely associate with GFAP-positive glial fibers across the inner nuclear layer. They also express markers of multipotency, like Pax6 and Sox2, as well as markers of neurogenesis, like Atoh7 (*Fausett and Goldman, 2006*; *Gorsuch et al., 2017*; *Nagashima et al., 2013*; *Raymond et al., 2006*; *Thummel et al., 2010*). Previous studies have investigated RPCs molecular identity mainly at the population level (*Cameron et al., 2005*; *Hoang et al., 2020*; *Kassen et al., 2007*; *Kramer et al., 2021*; *Lahne et al., 2020a*; *Morris et al., 2011*; *Ng Chi Kei et al., 2017*; *Powell et al., 2016*; *Qin et al., 2009*; *Ramachandran et al., 2012*; *Thummel et al., 2008*). However, a detailed characterization of the transcriptional network underlying RPC production and RPC differentiation, as well as the lineage relationships between MG, RPCs and differentiating progeny is currently missing. Here, we profiled the transcriptome of MG and MG-derived RPCs in response to an intense light lesion, and followed their differentiation trajectories towards regenerated retinal neurons at the single cell level. In particular, we addressed the transcriptional state of reactive MG prior to and after their asymmetric cell division. Moreover, we addressed whether MG-derived RPCs are heterogeneous and to what extent they express molecular markers that are observed during embryonic retinogenesis (*Xu et al., 2020*). To this aim, we employed a short-term lineage tracing strategy that allowed

the isolation of individual MG and MG-derived cells of the regenerating zebrafish retina, which were subsequently subjected to droplet-based, 10 x Genomics single-cell RNA sequencing (scRNAseq). We find that MG can be non-reactive in uninjured and lesioned conditions, as well as reactive upon lesion. Reactive MG give rise to a hybrid cell population, which replenishes the pool of MG as well as feeds into the pool of neurogenic RPCs. Moreover, MG-derived RPCs are highly heterogeneous and display a progressive restriction in their competence during subsequent differentiation. They upregulate developmental markers of retinogenesis and produce regenerated neurons partially according to the developmental birth order, with retinal ganglion cells born first and bipolar cells born last.

## Results

### Isolation of individual MG, RPCs and regenerated progeny using fluorescent reporters

Previous work on the regenerating zebrafish retina described the early phases of MG de-differentiation and RPC production (*Cameron et al., 2005*; *Fausett et al., 2008*; *Fausett and Goldman, 2006*; *Lahne et al., 2015*; *Nagashima et al., 2013*; *Nelson et al., 2013*; *Thummel et al., 2010*). However, knowledge on the molecular identity and cell fate choices of RPCs is scarce and based on analyses at a population level (*Cameron et al., 2005*; *Hoang et al., 2020*; *Kramer et al., 2021*; *Lahne et al., 2020a*; *Ng Chi Kei et al., 2017*; *Powell et al., 2016*; *Qin et al., 2009*; *Ramachandran et al., 2012*; *Thummel et al., 2008*). To study MG activation, RPC identity and their relationship with MG in more detail, we profiled the transcriptome of MG and MG-derived RPCs and followed their differentiation trajectories towards regenerated retinal neurons at the single cell level. To do so, we devised a strategy to isolate individual MG, RPCs and regenerated progeny, using retinae of uninjured and light-lesioned fish carrying the two fluorescent reporters *pcna:EGFP* and *gfap:mCherry*. The former drives the expression of EGFP under the regulatory elements of *proliferating cell nuclear antigen* (*pcna*), a known marker of proliferating cells, and thereby labels actively dividing cells, and, due to the persistence of EGFP, their short-term progeny. The other reporter drives expression of mCherry under the regulatory elements of *glial fibrillary acidic protein* (*gfap*), a known marker of zebrafish MG. Both transgenes recapitulate the endogenous expression in the adult zebrafish retina faithfully (*Figure 1—figure supplement 1*). Immunohistochemistry against the proliferation marker PCNA co-labeled *pcna:EGFP*-positive cells mainly in the ciliary marginal zone, one of the stem cells niches in the adult zebrafish retina (*Bernardos et al., 2007*; *Raymond et al., 2006*). Occasionally, *pcna:EGFP*/PCNA-double positive cells were observed in the central retina as well (*Figure 1—figure supplement 1A*). The MG-specific antibody Zrf1 labeled virtually all *gfap:mCherry*-positive cells in the central retina as well as in the ciliary marginal zone (*Bernardos et al., 2007*; *Figure 1—figure supplement 1B*). We also addressed the expression of the two fluorescent reporter lines following light lesion, at three different time points upon injury. First, 44 hours post light lesion (hpl), when MG are about to undergo or have already completed their asymmetric cell division giving rise to the first RPCs and self-renewed MG (*Lahne et al., 2015*; *Nagashima et al., 2013*). Second, 4 days post light lesion (dpl), representing the peak of RPC proliferation (*Bernardos et al., 2007*; *Fausett and Goldman, 2006*) and third, 6 dpl, when the first differentiating, regenerated progeny arise (*Bernardos et al., 2007*). Immunohistochemistry against EGFP and mCherry in double transgenic *Tg(pcna:EGFP);Tg(gfap:mCherry)* animals corroborated the expected expression pattern of the transgenes in the central retina at the chosen time points (*Figure 1—figure supplement 2*). Uninjured, central retinae displayed no EGFP, but strong mCherry signal consistent with non-proliferating MG (white arrowheads). In contrast, numerous EGFP/mCherry double-positive cells were present at 44 hpl, indicating potential cell-cycle re-entry of MG (yellow arrowheads). At 4 dpl, overall mCherry expression was reduced, consistent with the previously described downregulation of the *gfap* promoter by MG-derived RPCs (*Fimbel et al., 2007*; *Lenkowski and Raymond, 2014*; *Thummel et al., 2008*). In contrast, EGFP was strongly found throughout all retinal layers, matching the reported peak in proliferation of MG-derived RPCs (red arrowheads) (*Bernardos et al., 2007*; *Fausett and Goldman, 2006*; *Fimbel et al., 2007*; *Lenkowski and Raymond, 2014*; *Thummel et al., 2008*). Finally, in comparison to 4 dpl, an increase in mCherry accompanies a decline in EGFP signal at 6 dpl (yellow arrowhead). In order to obtain single cells of MG, RPCs and regenerated progeny, we utilized the persistence of EGFP and mCherry fluorescence, providing a short-term label. Initially, only MG express mCherry in uninjured,

central retinae, but activate EGFP in preparation of the asymmetric cell division upon light lesion at 44 hpl (*Figure 1A*). Firstborn RPCs inherit high levels of persisting fluorescent mCherry protein from dividing MG and slowly degrade/dilute the protein during subsequent rounds of cell division. Hence, MG-derived, but further dividing, RPCs are EGFP-positive, but low mCherry-positive at 4 dpl. At 6 dpl, RPCs have further diluted the mCherry reporter, thus displaying only a faint mCherry signal, but remain EGFP-positive. Similarly, RPC-derived, regenerated progeny is characterized by even lower levels of intrinsic mCherry fluorescence, and low EGFP signal during subsequent cell differentiation at 6 dpl. Following this rationale, we dissected retinae of uninjured and light-lesioned samples, dissociated them into single cell suspensions that were subjected to flow cytometry followed by droplet-based, 10 x Genomics scRNAseq (*Figure 1B*). Flow cytometry plots confirmed the rationale of the above-mentioned strategy (*Figure 1C* and *Figure 1—figure supplement 3*). *Figure 1C* depicts contour plots, which in flow cytometry represent the relative frequency of events, that is cells, in a sample. Each contour line in the plot encloses an equal percentage of cells, and contour lines that are closely packed indicate a high concentration of cells. In all samples, we detected non-fluorescent and EGFP-positive only cells, consistent with non-proliferating and proliferating cells in the adult zebrafish retina. We also detected cell populations expressing only mCherry at all sampled time points, most likely representing non-reactive MG. In addition to cells expressing either EGFP or mCherry, we find EGFP/mCherry-double positive cells in all samples consistent with proliferating MG originating from the ciliary marginal zone and the central retina (see *Figure 1—figure supplements 1 and 2*). The EGFP/mCherry-double positive population increased at 44 hpl, presumably due to an increase in the cell number of reactive, proliferating MG. A second, EGFP-positive, lower mCherry-positive cell population was detected at 44 hpl, presumably including the very first RPCs resulting from MG cell division at this time point. At 4 dpl, the large EGFP/mCherry-double positive cell population was still present, but displayed a lower mCherry, yet an increased EGFP signal, in line with MG-derived RPCs, which exhibit maximal proliferation at 4 dpl (*Bernardos et al., 2007*; *Fausett and Goldman, 2006*). At 6 dpl, a further decreased mCherry intensity was observed for the EGFP/mCherry-double positive cell population. In order to profile the transcriptome of individual MG, RPCs and progeny of the uninjured and light lesioned retinae, we dissected retinae of the before-mentioned time points (uninjured, 44 hpl, 4 dpl, 6 dpl) and eventually sorted 15,000 cells per time point. We applied a sorting gate that included both mCherry-positive only and EGFP/mCherry-double positive cells at the sampled time points (*Figure 1C* and *Figure 1—figure supplement 3*). Non-fluorescent as well as EGFP- only positive cells were not included. Taken together, we devised a short-term lineage tracing strategy that allows the isolation of individual MG and MG-derived cells of the regenerating zebrafish retina, which can be subjected to droplet-based,10x Genomics.

## scRNAseq identifies 15 distinct cell populations in the light-lesioned retina

Sorted cells underwent 10 x Genomics scRNAseq and 13,139 cells passed the sequencing quality control. We detected a median of 1.407 genes and 4.061 mean read counts per cell. Initially, we identified three clusters for the full dataset (*Figure 2—figure supplement 1A*): a retinal core dataset and two additional clusters identified as microglia and ribosomal gene-enriched cells (*Figure 2—figure supplement 1B*). Both populations were excluded from the subsequent analysis. Re-clustering of the core dataset, which eventually included 11,690 cells, revealed 15 potential, different cell populations (*Figure 2A*). From left to right on the UMAP, which visualizes the cell lineage trajectory, we observed the progressive emergence of MG, RPC and progeny clusters. We assigned the cellular identity by inspecting the top 100, upregulated marker genes per cluster (*Supplementary file 1*). In detail, we identified four MG clusters (clusters 1–4) that locate to the left and middle portion of the map, followed by five RPC clusters (clusters 5–9) locating to the right side of the map. Regenerated progeny branch from the RPCs and include retinal ganglion (cluster 10), amacrine (cluster 13) and bipolar cells (cluster 15) at the bottom-right, as well as red and blue cones (clusters 11 and 12) at the top-right of the map. Additionally, there is a second cluster of amacrine cells (cluster 14) that separates from the first amacrine cell cluster.

To investigate how the composition of cells changed during regeneration, we computed embedding densities for each time point (*Figure 2B*). Similar to the progression of cellular identities, cells progressed over time from left to right, from the uninjured, to the 44 hpl, the 4 dpl, as well as the 6

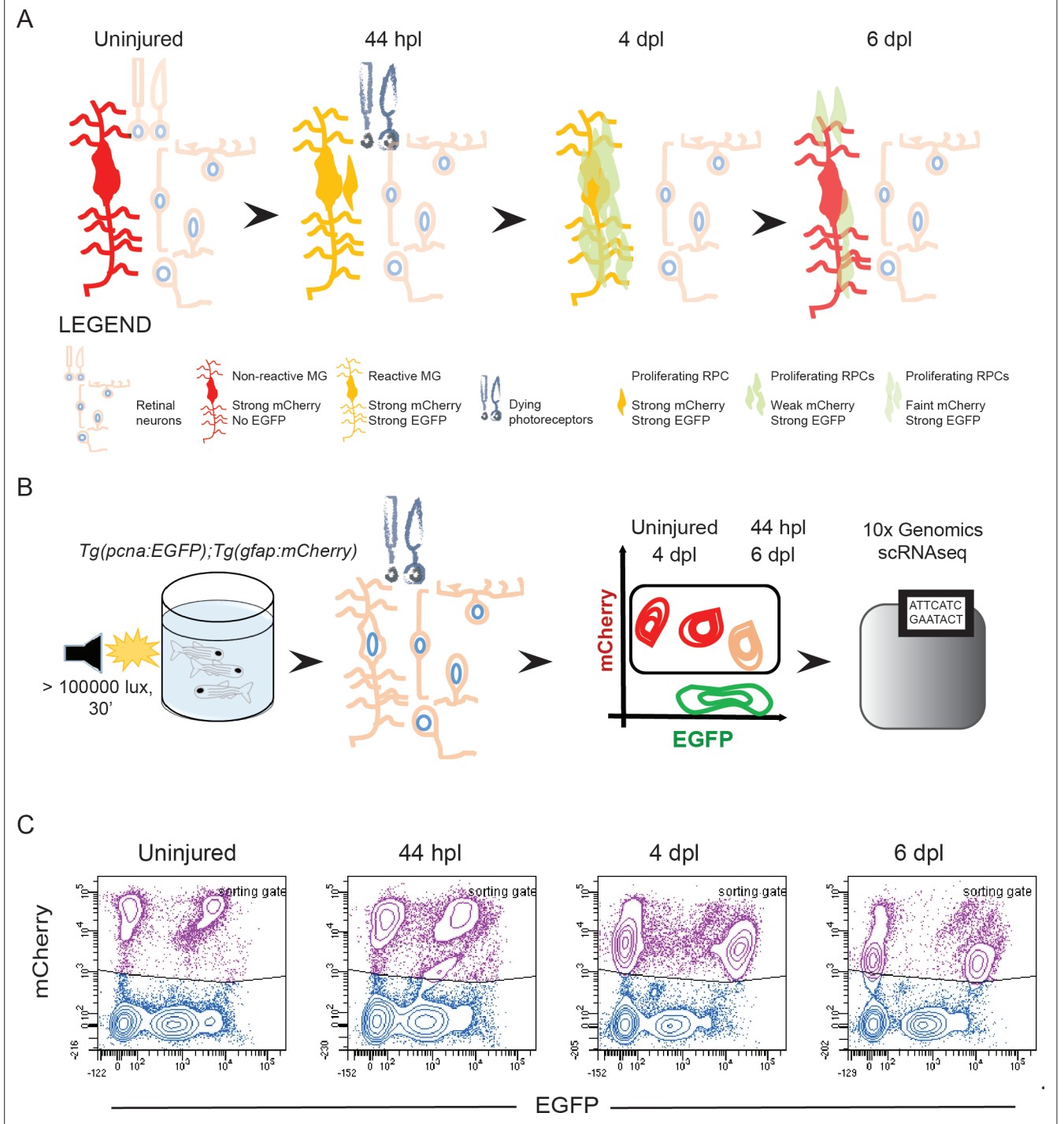

**Figure 1.** Isolation of individual MG, RPCs and regenerated progeny using fluorescent reporters. (**A**) Schematic illustration of the rationale to isolate individual cells from regenerating retinae of double transgenic *Tg(pcna:EGFP);Tg(gfap:mCherry)* animals. In the uninjured retina, MG express strong levels of mCherry but no EGFP. Upon light lesion, which results in dying photoreceptors, MG turn on EGFP resulting in the presence of both fluorophores at 44 hours post lesion (hpl). At 4 days post lesion (dpl), MG remain to contain strong mCherry and EGFP levels. In contrast, further proliferating RPCs display weak mCherry but strong EGFP fluorescence. Finally, MG have lost the EGFP signal at 6 dpl, expressing mCherry only. RPCs, which start to migrate and differentiate, are still detectable but show diluted levels of mCherry and EGFP. (**B**) Schematic illustration of the workflow for the scRNAseq experiment. Uninjured control and light lesioned retinae (44 hpl, 4 dpl, 6 dpl) of double transgenic *Tg(pcna:EGFP);Tg(gfap:mCherry)* animals were dissociated to obtain a single cell suspension. Following fluorescent activated cell sorting of mCherry-positive only and EGFP/mCherry-double positive cells, individual cells were eventually subjected to droplet-based, 10 x Genomics scRNAseq. (**C**) Flow cytometry plots depicting EGFP and mCherry fluorescence of dissociated retinal cells at the indicated time points. Cells in the sorting gate include mCherry-positive only as well as

*Figure 1 continued on next page*

*Figure 1 continued*

mCherry/EGFP-double positive cells. Number of cells sorted from the sorting gate per time point: 15,000. Associated: *Figure 1—figure supplements 1–3*.

The online version of this article includes the following figure supplement(s) for figure 1:

**Figure supplement 1.** characterization of the fluorescent reporter lines *Tg(pcna:EGFP)* and *Tg(gfap:mCherry)* in double transgenic animals.

**Figure supplement 2.** expression profile of *Tg(pcna:EGFP);Tg(gfap:mCherry)* during retina regeneration.

**Figure supplement 3.** fluorescence activated cell sorting (FACS) and gating strategies.

dpl enriched samples. As expected, cells belonging to non-reactive MG were found at all sampled time points. In particular, the vast majority of cells of the uninjured sample belonged to cluster 1 as well as cluster 2. However, we also observed a small proportion of cells (*Figure 2B*) aligning in a trajectory from non-reactive MG (left side of the map) through reactive MG (middle part) to RPCs and differentiating progeny (right side) in the uninjured sample. At 44 hpl, cells mapping to non-reactive MG were still abundantly present, although mostly shifted right towards a cluster representing reactive MG (cluster 3). Cells belonging to this cluster of reactive MG were also highly enriched in this sample. In contrast, cells mapping to the second cluster of reactive MG (cluster 4) as well as to RPCs (clusters 5–9) were overrepresented at 4 dpl, in line with the reported peak in proliferation at this time point (*Bernardos et al., 2007*; *Fausett and Goldman, 2006*). Cells mapping to RPCs were still frequently present at 6 dpl. Moreover, cells belonging to regenerated progeny (clusters 10–15), which first appear at 4 dpl, became predominantly enriched at 6 dpl. In addition to the overall progression from left to right, we noted that cells mapping to reactive MG of the second cluster (cluster 4) and the abutting portion of non-reactive MG (cluster 1) were also enriched at this time point, indicating a reversed direction.

Analysis of the cell cycle state further supported our cluster annotation. Specifically, known cell cycle marker genes (given in materials and methods) were used to calculate the predicted cell cycle state (*Macosko et al., 2015*; *Tirosh et al., 2016*; *Whitfield et al., 2002*), and the cell cycle phase (G1, S, G2/M) was projected onto the UMAP of the single cell transcriptome (*Figure 2C*). Differentiating or differentiated cells were considered as cells in G1 (*Wu et al., 2021*). Consistently, we observed that most cells of the left side and lower middle part as well as cells branching from the right side of the map were in G1, which comprised most non-reactive MG (clusters 1 and 2), one cluster of reactive MG (cluster 4), one RPC population (cluster 7) and the regenerated progeny (clusters 10–15). Cells in S phase were also present on the left side of the map, which represent non-reactive MG (clusters 1 and 2), but were in particular enriched in two domains of the middle part corresponding to reactive MG (cluster 3) and RPCs (clusters 5 and 7). In addition, two domains on the right side of the map showed an accumulation of cells in S phase, which include photoreceptor and horizontal cell precursors (clusters 8 and 9). The pattern of cells in G2/M phase complemented the pattern of cells in G1 and S phase. Cell proliferation pattern was further confirmed by *pcna* expression (*Figure 2D*). When plotting *pcna* expression onto the main UMAP of the single cell transcriptome, we observed high counts in one population of reactive MG (cluster 3) and in RPCs (clusters 5, 6, 8, and 9). In summary, scRNAseq analysis enabled the establishment of a transcriptome map, which reveals the progressive emergence of non-reactive MG, reactive MG and MG-derived progenitors and regenerated progeny.

## scRNAseq identifies two populations of non-reactive and two of reactive MG in the light-lesioned retina

MG in the homeostatic zebrafish retina are characterized by the expression of GFAP, as well as other canonical glial markers, like *apoeb*, *glula*, *cahz*, *rlbp1a* (*Bernardos and Raymond, 2006*; *Peterson et al., 2001*; *Raymond et al., 2006*; *Thummel et al., 2008*; *Yurco and Cameron, 2005*). In our scRNAseq dataset, we found clusters 1 and 2 expressing *apoeb*, *glula*, *cahz*, *rlbp1a* as marker genes (*Figure 2—figure supplement 2* and *Supplementary file 1*). When plotting *apoeb* expression onto the UMAP, we observed high counts in clusters 1 and 2, as well as in clusters 3 and 4 (*Figure 3A*). The glial marker *gfap*, found among the top 100 upregulated marker genes for cluster 4 (*Supplementary file 1*), showed a similar pattern, with high counts in clusters 1–4, but displayed high counts in cluster 5 as well (*Figure 3A*). Given the expression pattern of *apoeb* and *gfap*, we identified clusters 1–4 as MG populations. Further analysis indicated clusters 1 and 2 as non-reactive MG and clusters 3 and 4

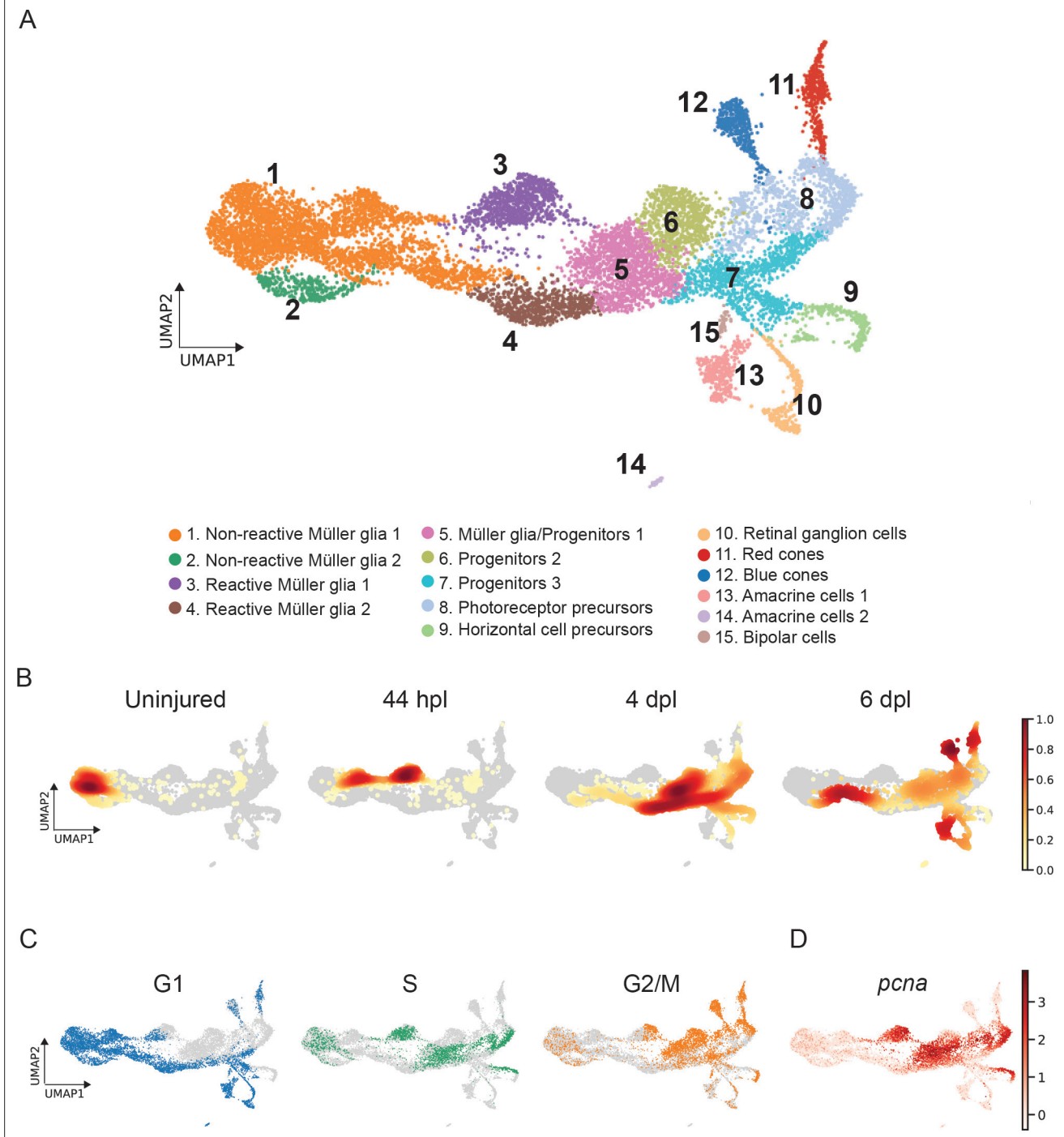

**Figure 2.** scRNAseq identifies 15 different cell populations in the light-lesioned retina. (**A**) A total of 11,690 cells depicted on the uniform manifold approximation and projection (UMAP) map. From left to right, the map shows a progressive emergence of MG, RPCs and progeny clusters. See figure legend for the detailed information about cluster identity. (**B**) Normalized cell density on the UMAP for the four different samples (uninjured, 44 hpl, 4 and 6 dpl). Dark red shows strong cell enrichment and no color absence of cells. (**C**) Cells colored by cell cycle phases (G1, S, and G2/M phase in blue, green and orange, respectively). (**D**) Expression of *pcna*. Legend: logarithm of normalized counts per million. Associated: *Figure 2—figure supplements 1 and 2*.

The online version of this article includes the following figure supplement(s) for figure 2:

**Figure supplement 1.** Original scRNAseq dataset.

**Figure supplement 2.** Dot plot of top 5 marker genes per cluster.

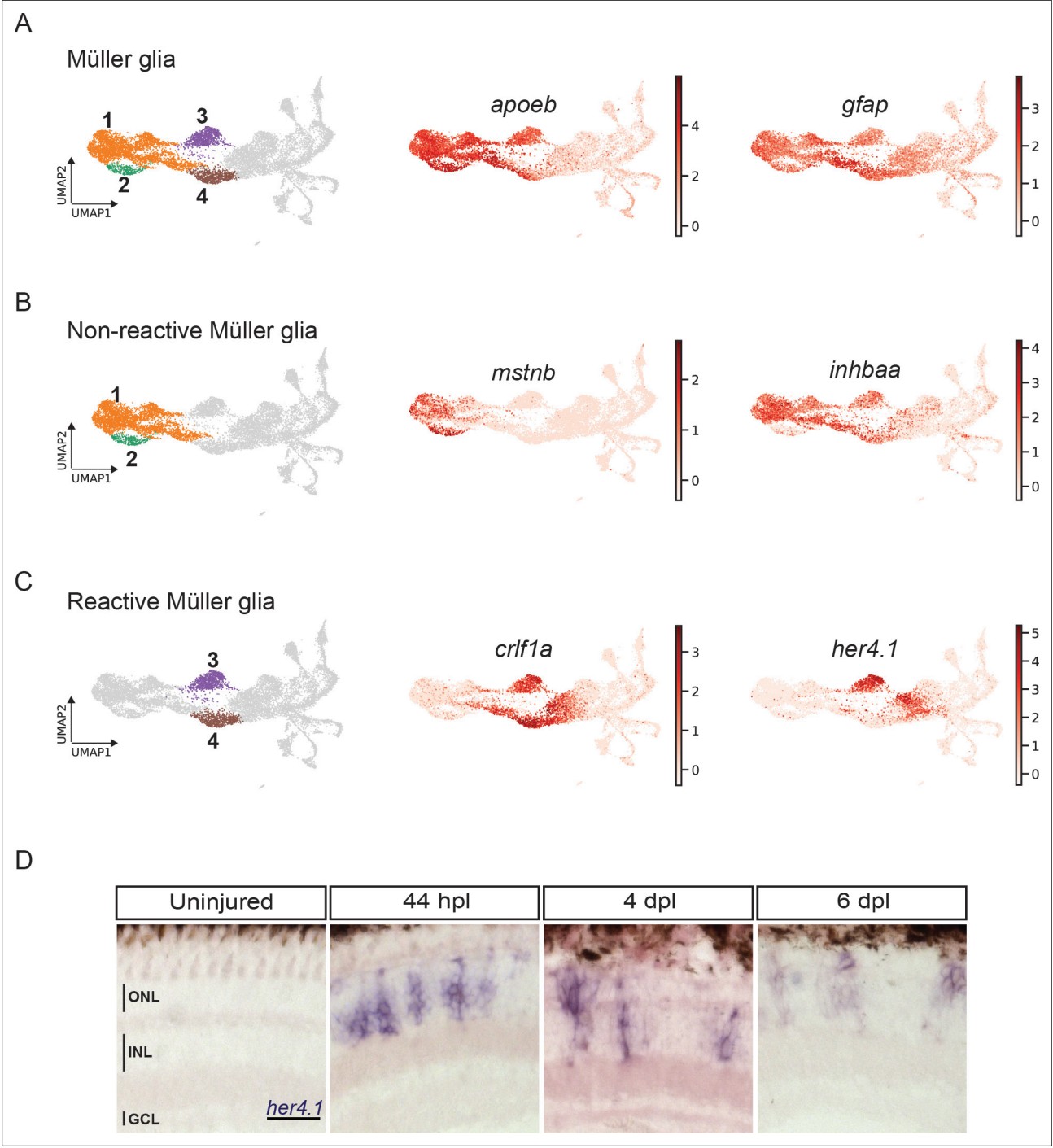

**Figure 3.** scRNAseq reveals heterogeneity in Müller glia and identifies two populations of non-reactive and two of reactive Müller glia in the light-lesioned retina. (**A**) Four Müller glia clusters (cluster 1–4) and the expression levels of *apoeb* and *gfap*. (**B**) The two non-reactive Müller glia clusters (cluster 1 and 2) and the expression levels of *mstnb* and *inhbaa*. Cells of both clusters express *mstnb*, but only cells in cluster 1 expresses *inhbaa*. (**C**) Visualization of the two reactive Müller glia clusters (cluster 3 and 4) and the relative expression levels of *crlf1a* and *her4.1* projected onto the main transcriptional UMAP. Cells of both clusters express *crlf1a*, but only cells in cluster 3 expresses *her4.1*. (**D**) Bright field microscopy images of chromogenic in situ hybridization detecting *her4.1* on retinal sections of uninjured control animals and at 44 hr post lesion (hpl), 4 days post lesion (dpl) and 6 dpl. N=3 fish per time point and uninjured controls. The position of the outer nuclear layer (ONL), inner nuclear layer (INL) and ganglion cell layer (GCL) is indicated in the uninjured sample. Scale bar: 30 µm. Associated: *Figure 3—figure supplements 1 and 2*.

The online version of this article includes the following figure supplement(s) for figure 3:

**Figure supplement 1.** expression of selected genes per cluster.

**Figure supplement 2.** expression profile of *stm* during retina regeneration.

as reactive MG (*Figure 3B and C*). Non-reactive MG, which were mostly in G1 and did not express any injury-specific markers (*Figure 2—figure supplement 2*) showed counts for *mstnb* (*Figure 3B*). In addition, members of the retinoic acid signalling pathway, like *rdh10a*, *aldh1a3*, *dhrs3a* (*D'Aniello et al., 2015*; *Feng et al., 2010*; *Holmes and Hempel, 2011*) were enriched in these clusters (*Figure 3—figure supplement 1A*, *Supplementary file 2*).

We next looked into differentially expressed genes (DEG) between the non-reactive MG clusters 1 and 2. For instance, the UMAP plot of *inhbaa*, an evolutionarily conserved vertebrate glia marker (*Hoang et al., 2020*) showed strong expression only in clusters 1, 3, and 4, but not in cluster 2 (*Figure 3B*, *Figure 3—figure supplement 1A*, and *Supplementary file 2*). Reactive MG on the other hand downregulated some glial markers, like *glula* and *rlbp1a*, and upregulated inflammatory cytokines, like *crlf1a*, matrix metalloproteases, like *mmp9*, and growth factors like *hbegfa* in the regenerating retina (*Figure 2—figure supplement 2*, *Figure 3—figure supplement 1B*, and *Supplementary file 1*). Consistent with that, plotting of *crlf1a* expression onto the UMAP showed high read counts in clusters 3 and 4, as well as in cluster 5 (*Figure 3C*). Again, DEG analysis showed that reactive MG can be subdivided further, because only cluster 3, but not cluster 4 expressed Hairy-related downstream targets of active Notch signalling including *her15.1–1*, *her9*, and *her4.1* (*Figure 2—figure supplement 2*, *Figure 3—figure supplement 1B*, and *Supplementary file 3*). Indeed, the UMAP plot of *her4.1* showed strong expression in cluster 3 as well as in cluster 5 (*Figure 3C*). To validate the expression of *her4.1* in the regenerating retina we used in situ hybridization. In agreement with our scRNAseq dataset, *her4.1* signal was not present in the uninjured retina, but appeared at 44 hpl in the inner nuclear layer of the central retina (*Figure 3D*). At 4 and 6 dpl, *her4.1* transcripts remained in the inner nuclear layer and appeared in cells of the outer nuclear layer too (*Figure 3D*). DEG analysis revealed *starmaker* (*stm*) as a highly specific marker for reactive MG in cluster 3, but not for reactive MG in cluster 4 (*Supplementary files 1 and 3*). We used fluorescent in situ hybridization to validate *stm* expression, combined with immunohistochemistry against GFAP (located mainly to MG fibers) as well as the proliferation marker PCNA in the uninjured as well as light-lesioned retina (*Figure 3—figure supplement 2*). In agreement with our scRNAseq results, we observed *stm* expression only at 44 hpl, but did not detect any *stm* expression in the uninjured as well as in the 4 and 6 dpl retinal samples (*Figure 3—figure supplement 2*). Virtually, all *stm*-positive cells at 44 hpl displayed a MG morphology (*Figure 3—figure supplement 2*, yellow arrowhead) and were positive for Zrf1 (detecting GFAP) as well as PCNA. We continued our validation by further analysing the expression of *crlf1a*, a marker of reactive MG, with fluorescent in situ hybridization combined with immunohistochemistry against GFAP and PCNA (*Figure 4*). Consistent with our scRNAseq dataset, *crlf1a* was not detectable in the uninjured, central retina, but was upregulated in Zrf1/PCNA-double positive cells at 44 hpl (*Figure 4A and B*). At 4 dpl, *crlf1a* signal was still present in Zrf1/PCNA-double positive cells and returned back to undetectable levels at 6 dpl (*Figure 4A and B*). Taken together, we observed cellular heterogeneity in MG during regeneration, and identified two non-reactive and two reactive MG populations that express distinct marker genes.

## scRNAseq identifies a transient cell population with hybrid characteristics of reactive MG and early RPCs in the light-lesioned retina

As mentioned above, several MG-related genes, like *gfap* or *crlf1a*, were also expressed in cluster 5, which locates next to the reactive MG (clusters 3 and 4), is highly proliferative and only becomes eminent at 4 dpl. Further analysis of this cluster showed expression of several members of the High Mobility Group (Hmg) protein family among the top 100 upregulated marker genes (*Supplementary file 1*). Consistent with that, the UMAP expression profile of *hmgb2b* showed presence of high counts already in cluster 3 (reactive MG), as well as in other, further downstream RPC clusters and regenerating progeny (*Figure 5A*). This indicated a transition from MG to RPC fate in the course of regeneration and that cells of cluster 5 might have hybrid characteristics of reactive MG and downstream RPCs. Hence, we conducted a DEG analysis of cluster 5 in comparison to reactive MG (clusters 3 and 4), as well as to the neighbouring, downstream RPCs (cluster 6). When compared to reactive MG, cells of cluster 5 showed expression of several genes, including *stmn1a*, *nr2e1* and *sox9b* (*Figure 3—figure supplement 1C*, *Supplementary files 4 and 5*). Consistently, UMAP expression profile of *stmn1a* showed counts prevalently in cluster 5 and neighbouring, downstream RPCs on the right side of the

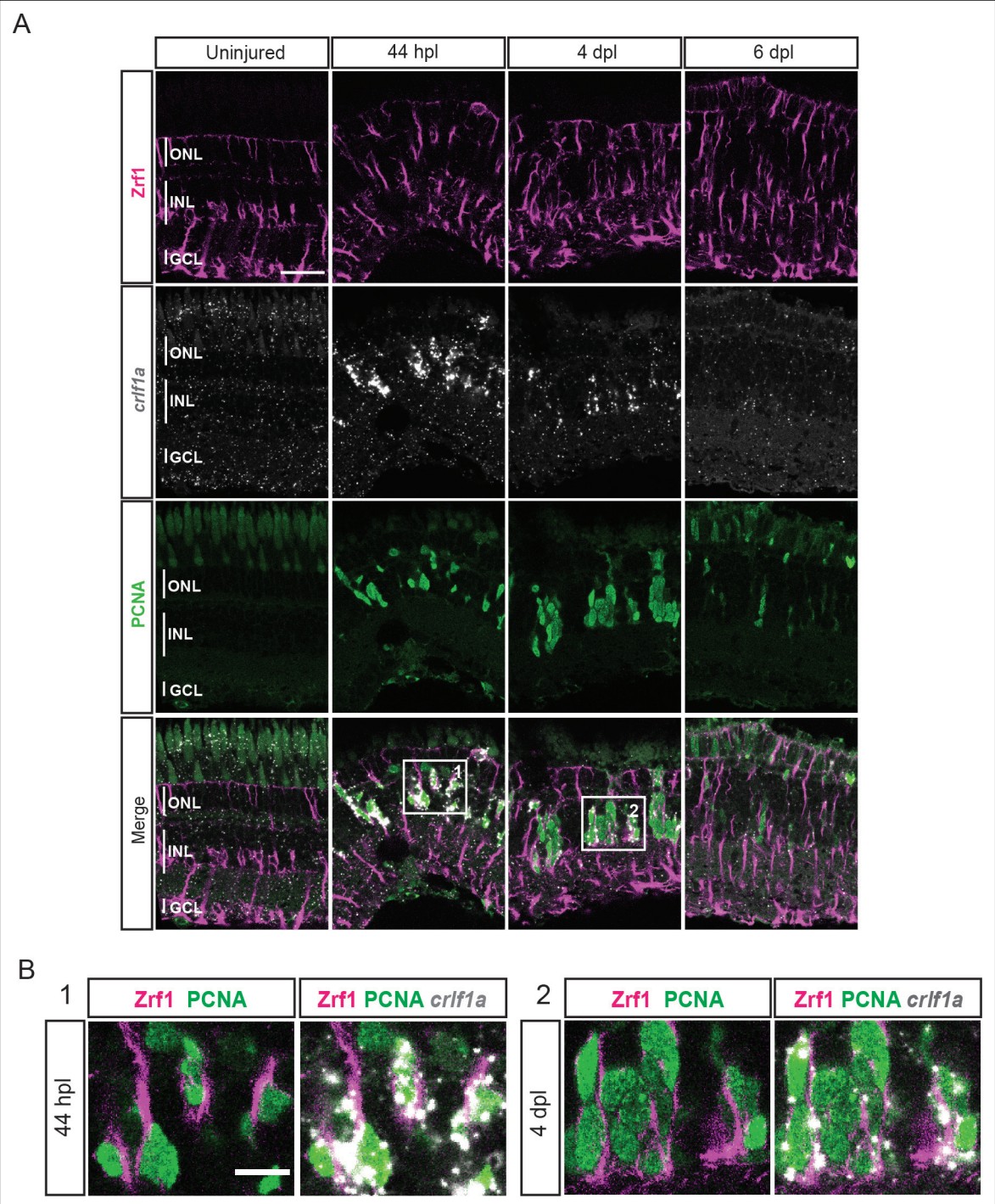

**Figure 4.** expression profile of *crlf1a* during retina regeneration. (**A**) Confocal images of fluorescent in situ hybridization for *crlf1a* (grey) and immunohistochemistry detecting Zrf1-positive MG fibers (magenta) and the proliferation marker PCNA (green) on retinal sections of uninjured control animals and at 44 hours post lesion (hpl), 4 days post lesion (dpl) and 6 dpl. The position of the outer nuclear layer (ONL), inner nuclear laser (INL) and ganglion cell layer (GCL) is indicated in the uninjured sample. N=3 fish per time point and uninjured controls. Scale bar: 30 μm. (**B**) Magnification of the areas indicated by the squares 1 and 2 in panel A. Scale bar: 10 μm.

map, but not in the reactive or non-reactive MG on the left side (*Figure 5A*). The UMAP profiles of *sox9b* and *nr2e1* showed strong expression almost exclusively restricted to cluster 5. DEG analysis between cluster 5 and the neighbouring downstream cluster 6 revealed also expression of *nr2e1* and *sox9b*, but additionally transcripts like *id1* (*Figure 3—figure supplement 1D* and *Supplementary*

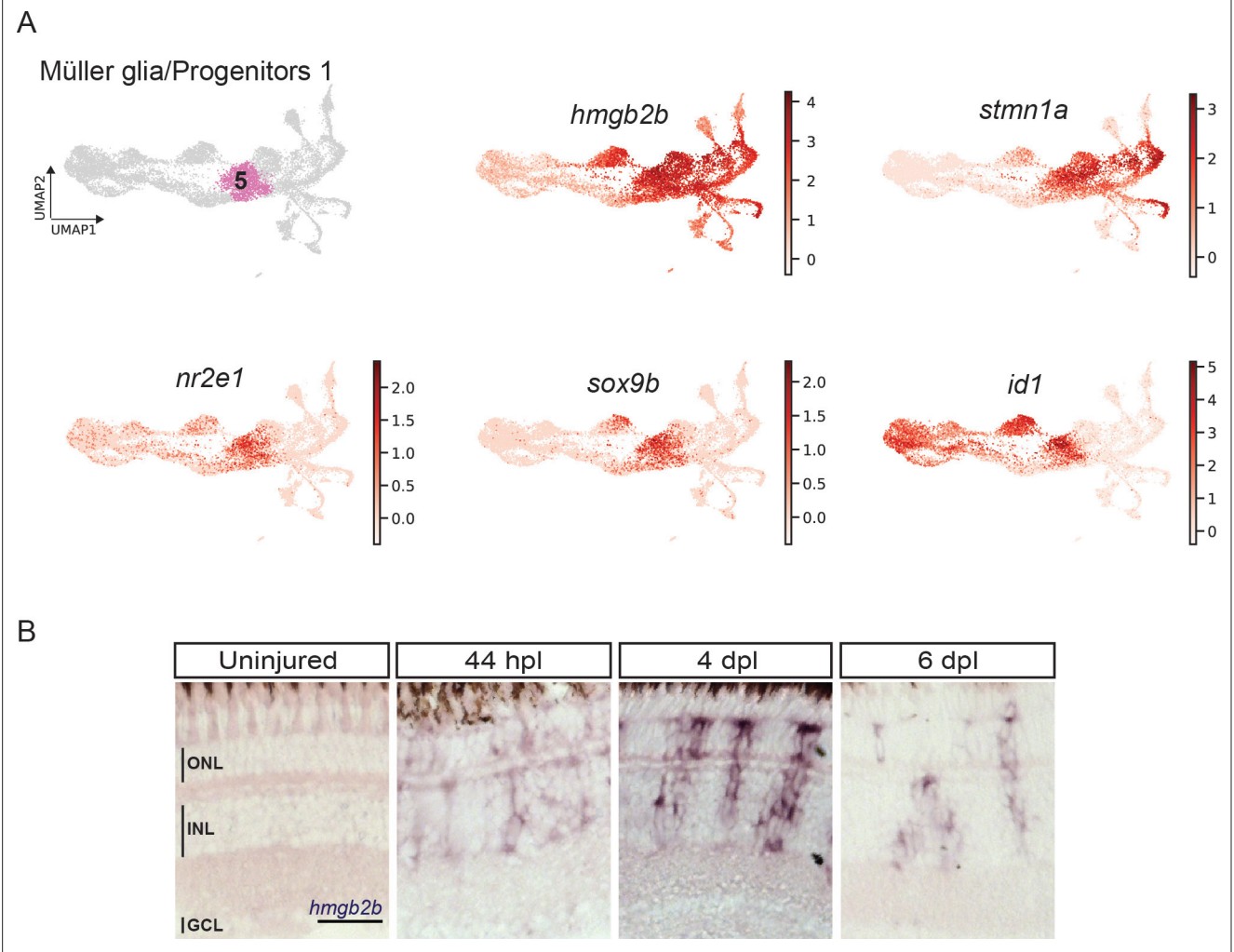

**Figure 5.** scRNAseq identifies a transient cell population with hybrid characteristics of reactive Müller glia and early progenitors in the light-lesioned retina. (**A**) The progenitor 1 cluster (cluster 5) and the expression levels of *hmgb2b*, *stmn1a*, *nr2e1 sox9b* and *id1*. (**B**) Bright field microscopy images of chromogenic in situ hybridization detecting *hmgb2b* on retinal sections of uninjured control animals and at 44 hours post lesion (hpl), 4 days post lesion (dpl) and 6 dpl. N=3 fish per time point and uninjured controls. The position of the outer nuclear layer (ONL), inner nuclear layer (INL) and ganglion cell layer (GCL) is indicated in the uninjured sample. Scale bar: 30 µm.

*file 6*). Consistently, high *id1* counts were found in cluster 5 and in other reactive or non-reactive MG clusters, but not in downstream RPCs on the right side of the map (*Figure 5A*). In order to validate the RNAseq data, we analysed the expression of *hmgb2b* and *id1* transcripts in the regenerating retina using in situ hybridization and fluorescent in situ hybridization combined with immunohisto-chemistry, respectively. Consistent with our scRNAseq data, *hmgb2b* signal was not detectable in the uninjured, central retina, but was initiated in individual cells residing in the inner nuclear layer at 44 hpl (*Figure 5B*). Subsequently, *hmgb2b* transcripts could be detected in clusters of cells in the inner and outer nuclear layers at 4 and 6 dpl (*Figure 5B*). The expression of *id1* in the regenerating tissue matched the scRNAseq data. Expression of *id1* was already detectable in Zrf1-positive but PCNA-negative cells in the inner nuclear layer of the uninjured, central retina (*Figure 6A and B*). At 44 hpl, *id1* was strongly upregulated and co-localized with Zrf1/PCNA-double positive cells in the inner nuclear layer. At 4 and 6 dpl, *id1* expression was still associated with Zrf1/PCNA-double positive cells and, occasionally, with PCNA-only positive cells (*Figure 6A and B*). In summary, marker analysis reveals that cells in cluster 5 display a transcriptional signature of reactive MG, as well as RPCs, which is why we assigned it a hybrid identity.

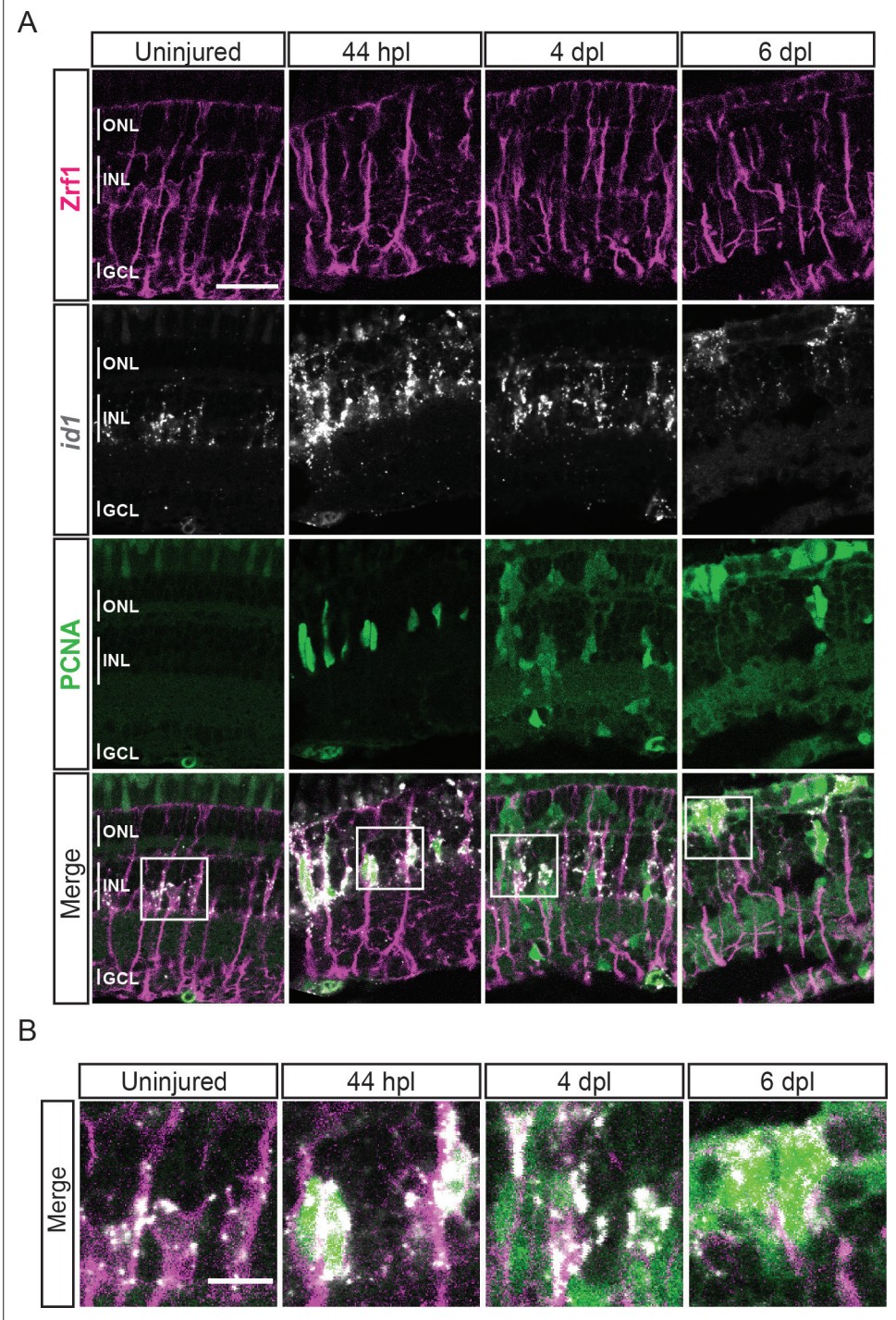

**Figure 6.** expression profile of *id1* during retina regeneration. (**A**) Confocal images of fluorescent in situ hybridization for *id1* (grey) and immunohistochemistry detecting Zrf1-positive MG fibers (magenta) and the proliferation marker PCNA (green) on retinal sections of uninjured control animals and at 44 hours post lesion (hpl), 4 and 6 days post lesion (dpl). The position of the outer nuclear layer (ONL), inner nuclear layer (INL) and ganglion cell layer (GCL) is indicated in the uninjured sample. N=3 fish per time point and uninjured controls. Scale bar: 30 μm. (**B**) Magnification of the areas indicated by the squares 1 and 2 in panel A. Scale bar: 10 μm.

## scRNAseq identifies two neurogenic progenitor populations in the light-lesioned retina

The before-mentioned DEG-analysis between cluster 5 (hybrid cells) and cluster 6 (neighbouring, downstream RPCs) also revealed that the latter upregulated *insm1a* (*Figure 3—figure supplement 1D* and *Supplementary file 6*), which has been described to promote cell cycle exit of RPCs in the injured fish retina at 4 dpl (*Ramachandran et al., 2012*). In addition, our DEG analysis showed upregulation of *atoh7* and *onecut2* in cluster 6 (*Figure 3—figure supplement 1D* and *Supplementary file 6*). We found that the highly redundant gene *onecut1* was also expressed in cluster 6, albeit this was not found in the DEG analysis. As expected, the UMAP expression profile of *insm1a* confirmed high read counts in cluster 6 but also in clusters 7, 8, and 10 (*Figure 7A*). The UMAP expression profile of *atoh7* also confirmed expression in cluster 6, as well as in clusters 7 and 10 (*Figure 7A*). Moreover, the UMAP expression profile of *onecut1* and *onecut2* confirmed that both genes were expressed in clusters 6 and 7 and in addition in cluster 9 (*Figure 7A*). Cluster 7 expressed additional neurogenic marker genes like *tubb5* and *sox11b* (*Baraban et al., 2013*; *Lange et al., 2020*; *Mu et al., 2017*; *Figure 3—figure supplement 1D* and *Supplementary file 1*). Finally, the cell cycle status and the UMAP expression of *mki67*, a mitotic marker, showed read counts in cluster 6, but not in cluster 7 (*Figure 7A*). Hence, we concluded that cluster 6 and cluster 7 represented neurogenic progenitors, and were named progenitors 2 and progenitors 3, respectively. To validate *atoh7*, *onecut1* and *onecut2* expression in the regenerating retina, we employed again in situ hybridization. In agreement with our scRNAseq data, *atoh7* signal was absent in the uninjured and 44 hpl retina, but became detectable in the inner and outer nuclear layers at 4 and 6 dpl (*Figure 7B*). Similarly, *onecut1* and *onecut2* were prominently expressed in the inner and outer nuclear layer at 4 and 6 dpl, in addition to their homeostatic expression in the presumptive horizontal cell layer in the uninjured and 44 hpl retina (*Figure 7B*, black arrowheads). In summary, scRNAseq showed heterogeneity of neurogenic progenitors, which upregulate known markers of retinogenesis in the regenerating retina.

## scRNAseq identifies fate-restricted progenitors in the light-lesioned retina

Gene expression analysis of cluster 8, which locates next to the neurogenic progenitors (cluster 6 and 7), showed significant upregulation of several well-known photoreceptor precursor markers, including *crx*, and *thrb* (*Supplementary file 1*). Consistently, UMAP plots for *crx* and *thrb* revealed high counts in cluster 8 (*Figure 8A*). Moreover, the former was also highly expressed in clusters 11 and 12, whereas the latter was restricted to the additional cluster 1. Subsequent further gene expression analysis identified clusters 11 and 12 as differentiating cone photoreceptors arising from cluster 8. Precisely, clusters 11 and 12 represent differentiating photoreceptors giving rise to blue and red cones as shown by the UMAP plots of *opn1sw2* and *opn1lw2*, respectively (*Figure 8B*). Cluster 9, which abuts the neurogenic progenitor 3 (cluster 7), expressed known developmental markers of horizontal cell precursors like *prox1a* and *rem1* (*Supplementary file 1*). Consistently, the UMAP plot for *rem1* showed high expression almost restricted to this cluster specifically. Gene expression analysis also revealed *ptf1a* expression in cluster 9, which has been shown to be required for lineage specification of horizontal and amacrine cells in the developing retina (*Jusuf et al., 2011*; *Jusuf and Harris, 2009*). In line with these findings, the UMAP expression profile of *ptf1a* confirms high counts in the left part of cluster 9, but is even more pronounced in the upstream part of cluster 7 feeding into cluster 9 (*Figure 8C*). Gene expression analysis of the remaining clusters 10, 13, 14, and 15 identified them as differentiating progeny giving rise to retinal ganglion cells (cluster 10), amacrine cells (clusters 13 and 14), as well as bipolar cells (cluster 15), respectively (*Supplementary file 1*). In this context, UMAP plots of *gap43* and the more mature retinal ganglion cell marker *isl2b* showed a strong upregulation in cluster 10 (*Figure 8D*). UMAP plots of *stx1b* and *pvalb6*, markers of differentiating and mature amacrine cells, revealed strong expression in clusters 13 and 14, whereas UMAP plots of *vsx1* and *samsn1a*, known and novel markers of bipolar cells, respectively, showed high counts in cluster 15 (*Figure 8E and F*).

With this respect, it was recently shown that the regenerating adult zebrafish retina recapitulates developmental fate specification programs (*Lahne et al., 2020a*). Hence, we plotted the fraction of cells that differentiate into a specific cell fate during the time course of regeneration (*Figure 8—figure supplement 1*). We found that the earliest committed progeny emerged at 4 dpl giving rise to retinal ganglion cells or to photoreceptors, in particular red cones. At 6 dpl, the number of retinal ganglion

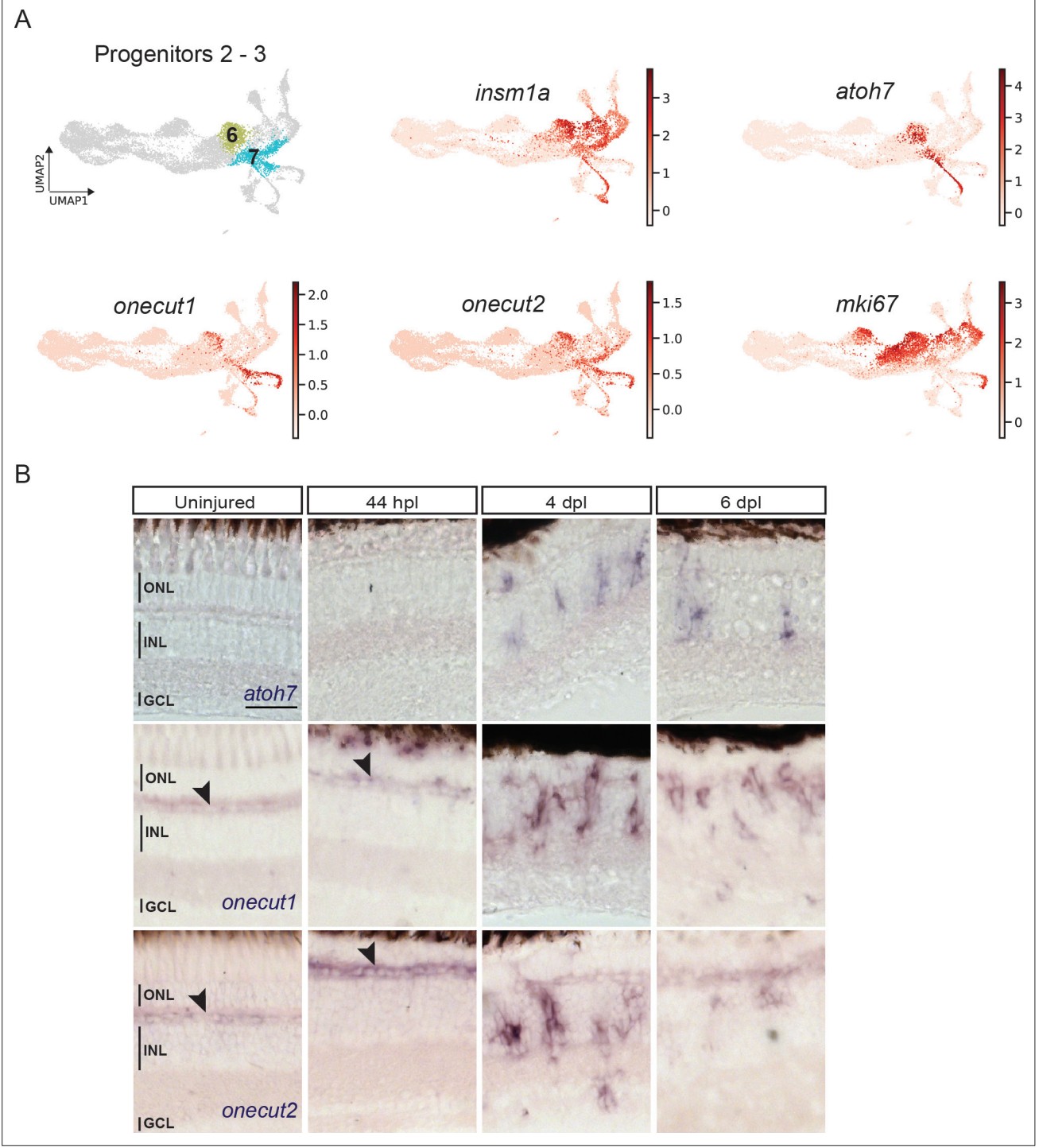

**Figure 7.** scRNAseq identifies two neurogenic progenitor populations in the light-lesioned retina. (**A**) The two progenitor clusters (clusters 6 and 7) and the relative expression levels of *insm1a*, *atoh7*, *onecut1*, *onecut2* and *mki67*. (**B**) Bright field microscopy images of chromogenic in situ hybridization detecting *atoh7*, *onecut1* or *onecut2* on retinal sections of uninjured control animals and at 44 hours post lesion (hpl), 4 days post lesion (dpl) and 6 dpl. Each ISH was carried on N=3 fish per time point and uninjured controls.The position of the outer nuclear layer (ONL), inner nuclear layer (INL), and ganglion cell layer (GCL) is indicated in the uninjured sample. Scale bar: 30 μm.

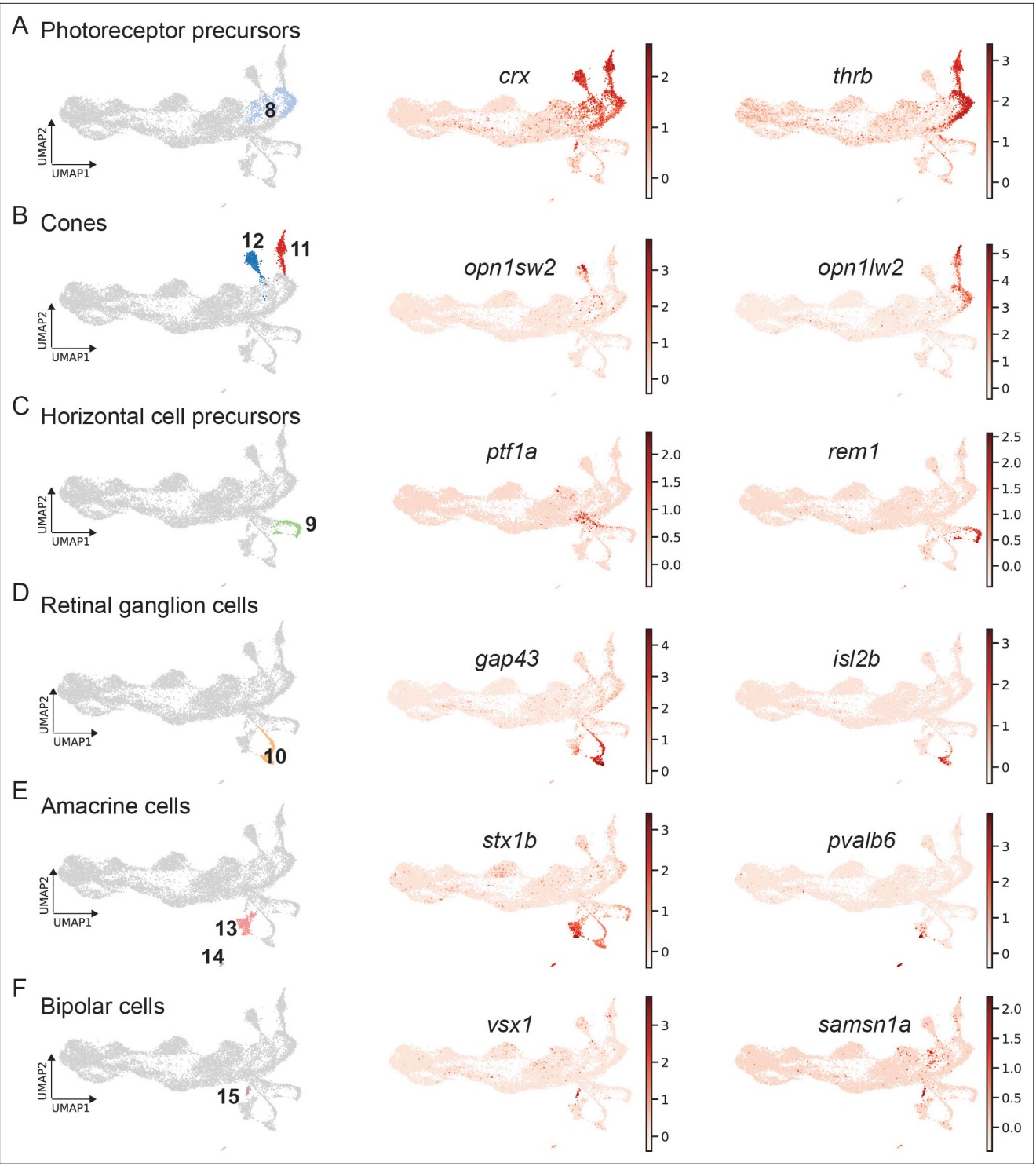

**Figure 8.** scRNAseq identifies fate-restricted progenitors and differentiating progeny in the light-lesioned retina. (**A**) The photoreceptor precursor cluster (cluster 8) and the expression levels of *crx* and *thrb*. (**B**) The red and blue cone cluster (clusters 11 and 12) and the expression levels of the blue cone marker *opn1sw2* and red cone marker *opn1lw2*. (**C**) The horizontal cell precursor cluster (cluster 9) and the expression levels of *ptf1a* and *rem1*. (**D**) The ganglion cell cluster (cluster 10) and the expression levels of *gap43* and *isl2b*. (**E**) The two amacrine cell clusters (clusters 14 and 15) and the expression levels of *stx1b* and *pvalb6*. (**D**) The bipolar cell cluster (cluster 15) and the expression levels of the known bipolar cell marker *vsx1* and the novel marker *samsn1a*. Associated: *Figure 8—figure supplements 1 and 2*.

The online version of this article includes the following figure supplement(s) for figure 8:

**Figure supplement 1.** birthdate order of regenerated progeny.

**Figure supplement 2.** expression of mature markers.

cells and red cones continued to increase. In addition, we detected a significant number of blue cones and amacrine cells, as well as a small number of bipolar cells at this time point. We did not detect any differentiating progeny giving rise to horizontal cells or rods at this stage. To address this issue, we further investigated the expression of more mature neuronal markers in our dataset (*Figure 8—figure supplement 2*). Expression of *rho* and *gnat1*, two markers of rod photoreceptors, was very low and sparse, and did not define any specific rod cluster (*Figure 8—figure supplement 2A*). Expression of *gad1b*, which encodes for glutamate decarboxylase synthetizing the neurotransmitter GABA, was mostly restricted to amacrine cells (clusters 13 and 14) (*Figure 8—figure supplement 2B*). To address the expression of more mature markers of amacrine and horizontal cells, we investigated *chata* (*choline O-acetyltransferase a*, an amacrine cell-specific marker), as well as *calb2a* and *calb2b* (*calbindin 2 a* and *calbindin 2b*, markers of amacrine and horizontal cells) (*Figure 8—figure supplement 2B, C*). As expected, high counts of *chata* were specifically found in amacrine cells (clusters 13 and 14). In particular, *chata* expression was detected only in cells located at the tip of cluster 13 and in all cells of cluster 14. *calb2a* was expressed in some reactive MG (cluster 3) and quite strongly in some amacrine cells (cluster 13). Importantly, we did not detect any *calb2a* and *calb2b* in horizontal cell precursors (cluster 9).

We also checked the expression of transcripts for variants of protein kinase C using *prkcab* and *prkcbb* (*protein kinase alpha b*, *protein kinase beta b*) to identify mature bipolar cells (*Figure 8—figure supplement 2D*). Counts for *prkcab* were sparsely found in some non-reactive and reactive MG (clusters 1–4). *prkcbb*-positive cells were found more frequently in some non-reactive and reactive MG (clusters 1–4), progenitor 1 (cluster 5), as well as in amacrine cells (clusters 13, 14). Importantly, we did not detect *prkcab* or *prkcbb* in our bipolar cells. When plotting the expression of general synaptic markers like *snap25a* and *snap25b*, we observed mainly transcripts in the differentiating progeny (*Figure 8—figure supplement 2E*). Specifically, high counts of *snap25a* were found in retinal ganglion and amacrine cells (clusters 10, 13, and 14). High counts of *snap25b* were observed in the same clusters as well as in red and blue cones (clusters 11 and 12) and the right part of the horizontal cell precursors (cluster 9). Taken together, our scRNAseq data reveal the presence of fate-restricted progenitors and the sequential appearance of differentiating progeny in the light-lesioned, adult zebrafish retina. Moreover, the analysis of more mature markers shows that the differentiation of retinal ganglion cells and red cones is preceding the appearance of blue cones, amacrine and bipolar cells.

## scRNAseq of MG and MG-derived cells of the regenerating zebrafish retina corroborates and complements previously published scRNAseq data

In order to compare our newly obtained scRNAseq dataset of MG and MG-derived cells of the regenerating zebrafish retina to a previously published dataset of the light-lesioned retina (*Hoang et al., 2020*), we employed the ingestion method (Scanpy) and mapped the clusters identified by Hoang and colleagues to our clusters (*Figure 9*). While we applied a short-term lineage-tracing strategy and only sequenced the enriched population of FAC-sorted MG and MG-derived cells of the regenerating zebrafish retina, Hoang and colleagues sequenced all retinal cells in the light-lesioned retina. Consequently, comparison between the two datasets unravelled similarities, but also significant differences due to the different experimental set-ups (*Figure 9A*). Consistently, the cluster annotated as resting MG in Hoang et al. mapped to clusters annotated as non-reactive MG 1 and 2 in our dataset (*Figure 9B*). The cluster annotated as activated MG in Hoang et al. mapped to clusters annotated as reactive MG 1 and 2, as well as to the cluster with hybrid identity of MG/progenitors in our dataset. Interestingly, some cells annotated as activated MG in Hoang et al. mapped also to neurogenic progenitor 2 and 3 clusters in our dataset (*Figure 9B*). The cluster annotated as progenitors in Hoang et al. mapped to the progenitor area in our dataset, which included neurogenic progenitors 2, 3 as well as photoreceptor and horizontal cell precursors (*Figure 9B*). Finally, retinal ganglion cells, cones, GABAergic amacrine cells and bipolar cells annotated in Hoang et al. perfectly mapped to retinal ganglion cells, cone, amacrine, and bipolar cells in our dataset (*Figure 9B*). While we did not detect a mature horizontal cell cluster, Hoang and colleagues annotated a horizontal cell cluster, which cells mapped to reactive MG 2, MG/progenitors 1 and part of progenitors 3 in our dataset (*Figure 9B*). Moreover, Hoang and colleagues annotated rod photoreceptors that mapped to progenitors 3, photoreceptor precursors, red and blue cones, horizontal cell precursors and bipolar cells in our dataset (*Figure 9B*).

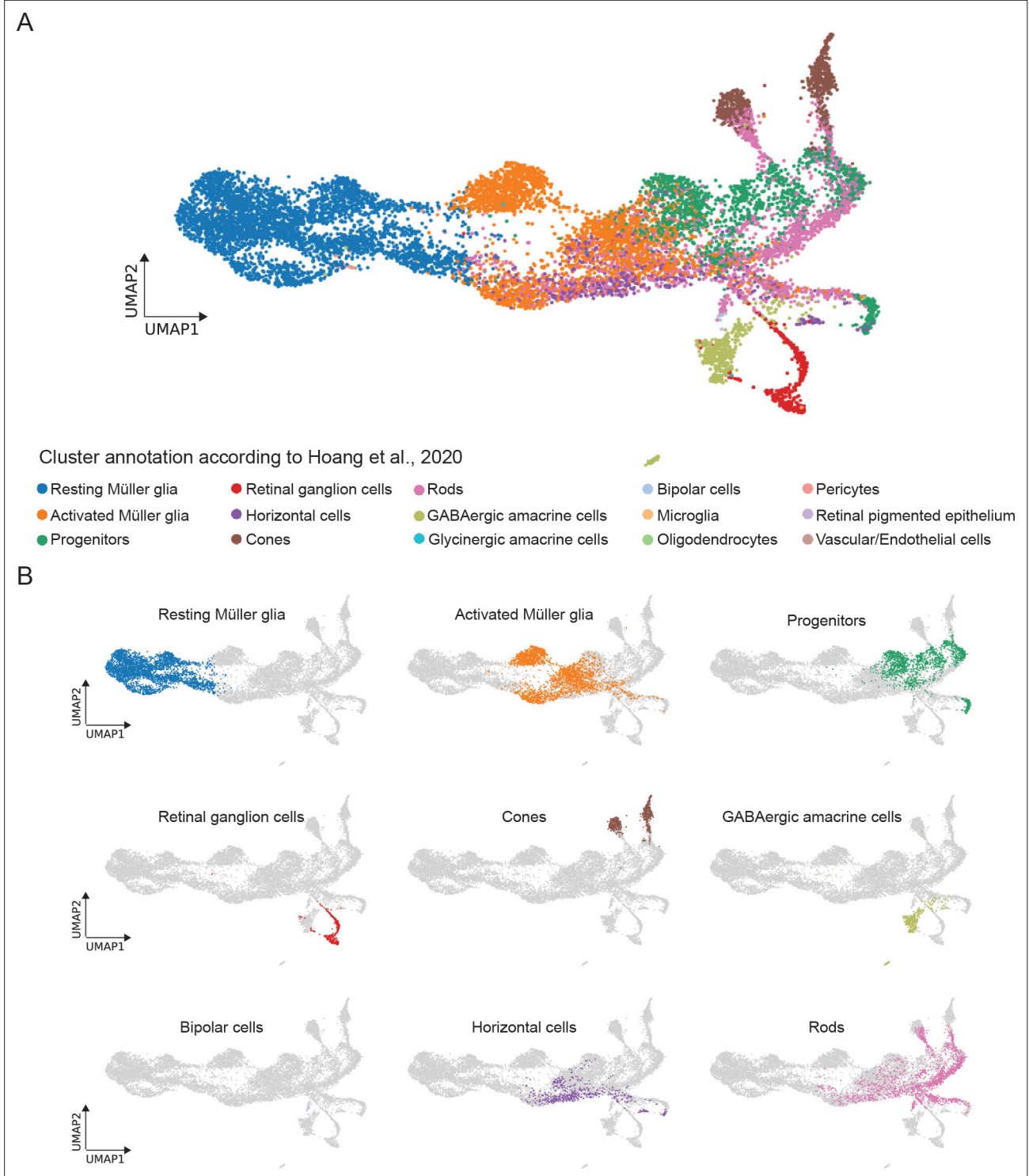

**Figure 9.** Comparison between the present scRNAseq dataset and the scRNAseq dataset by *Hoang et al., 2020*. (**A**) UMAP showing all cell clusters annotated in *Hoang et al., 2020* and mapped to the scRNAseq dataset described in the present study. (**B**) UMAP showing the individual cell clusters annotated in *Hoang et al., 2020* and mapped to the scRNAseq dataset described in the present study.

The online version of this article includes the following figure supplement(s) for figure 9:

**Figure supplement 1.** dot plot of selected marker genes per cluster in the present scRNAseq dataset and the scRNAseq dataset by *Hoang et al., 2020*.

Finally, Hoang and colleagues annotated additional cell clusters that did not map to any cluster in our dataset, and included oligodendrocytes, pericytes, retinal pigmented epithelial cells as well as vascular/endothelial cells (*Figure 9B*).

Next, we selected representative marker genes per cluster from our scRNAseq dataset and checked their expression in the dataset by Hoang and colleagues (*Figure 9—figure supplement 1*). The dot plot showing the expression of selected gene candidates per cluster further corroborated the large overlap between clusters annotated in the present study with those annotated in the study by Hoang and colleagues.

## Discussion

In this study, we employed a short-term lineage tracing strategy, which allowed the selective enrichment of MG and their respective progeny for up to six days post light lesion. Subsequent scRNAseq resolved the transcriptome of individual MG, RPCs as well as regenerated progeny, and allowed to reconstruct their lineage relationship in the homeostatic and regenerating zebrafish retina (*Figure 1*). A previous study combined bulk RNA sequencing of sorted MG with scRNAseq of whole retinae from zebrafish upon NMDA as well as light lesions (*Hoang et al., 2020*). This work focussed specifically on the early reprogramming of MG and progenitor production, covering up to 36 hr post light lesion. The authors report that MG go through an activated, non-proliferative state characterized by upregulation of gliosis markers. However, neither did they find any evidence of MG heterogeneity in uninjured conditions nor did they provide any thorough characterization of MG-derived RPCs (*Hoang et al., 2020*). Additional RNA sequencing studies of the regenerating zebrafish retina have used microarray as well as RT-PCR and qRT-PCR to evaluate transcriptional changes at a population level, without providing any MG/RPC single cell resolution (*Cameron et al., 2005*; *Craig et al., 2008*; *Kassen et al., 2007*; *Kramer et al., 2021*; *Morris et al., 2011*; *Ng Chi Kei et al., 2017*; *Qin et al., 2009*). Previous immunohistochemistry and in situ hybridization experiments provided a broad description of MG-derived RPCs. Indeed, progenitors are reported as fusiform cells that proliferate and express markers of multipotency like Pax6 and Sox2, as well as some neurogenic markers like Atoh7, Ptf1a, Vsx1 and others (*Bernardos et al., 2007*; *Fausett and Goldman, 2006*; *Fimbel et al., 2007*; *Gorsuch et al., 2017*; *Lahne et al., 2020a*; *Ramachandran et al., 2012*). In contrast to these previous results, our study is the first providing a comprehensive overview of MG, MG-derived cells and their lineage relationship up to 6 days post lesion in the zebrafish retina. Due to our selective enrichment of MG and their respective progeny, we gain an adequate single-cell resolution to observe substantial heterogeneity in the molecular identity of MG in homeostatic as well as regenerating conditions. We also provide an in depth description of MG as well as RPC molecular identity, cell fate decisions and production of regenerated progeny that cover all key aspects during zebrafish retina regeneration. Our strategy allowed us to define two main trajectories in the regenerating zebrafish retina (*Figure 10*). The first trajectory (solid and dashed red lines) comprises clusters 1–5 and constitutes a closed cycle that both maintains the MG pool and generates the first-born neurogenic RPCs. The second trajectory (solid black lines) involves clusters 6–15 and gives rise to all retinal neurons via the progressive restriction in cell fate competence of neurogenic RPCs. We discuss the two trajectories separately.

### Müller glia self-renew and give rise to early, multipotent progenitors

The glial trajectory starts with two populations of non-reactive MG (clusters 1 and 2), which express canonical glial markers like *gfap*, *apoeb* and additional genes like *glula*, *rlbp1a*, and *cahz* that ensure the housekeeping function of MG during homeostasis (*Figure 3A*, *Figure 2—figure supplement 2*, *Supplementary file 1*). The majority of cells in the non-reactive MG clusters of our scRNAseq dataset are quiescent based on the analysis of the cell cycle, and are frequently found in the uninjured, but also in the lesioned retina at all sampled time points (*Figure 2*). However, some non-reactive MG are in S or G2/M phases of the cell cycle, and express *pcna*, albeit at a lower level as compared to the expression in injury-induced, proliferating cell clusters (*Figure 2C and D*). This is consistent with the proposed model that zebrafish MG proliferate occasionally also in the uninjured, central retina and generate rod progenitors that feed the rod precursor pool at the base of the outer nuclear layer (*Bernardos et al., 2007*; *Stenkamp, 2007*). Accordingly, we detect PCNA-positive cells in the uninjured, central retina (*Figure 1—figure supplement 1*) as well as a small fraction of cells that align in

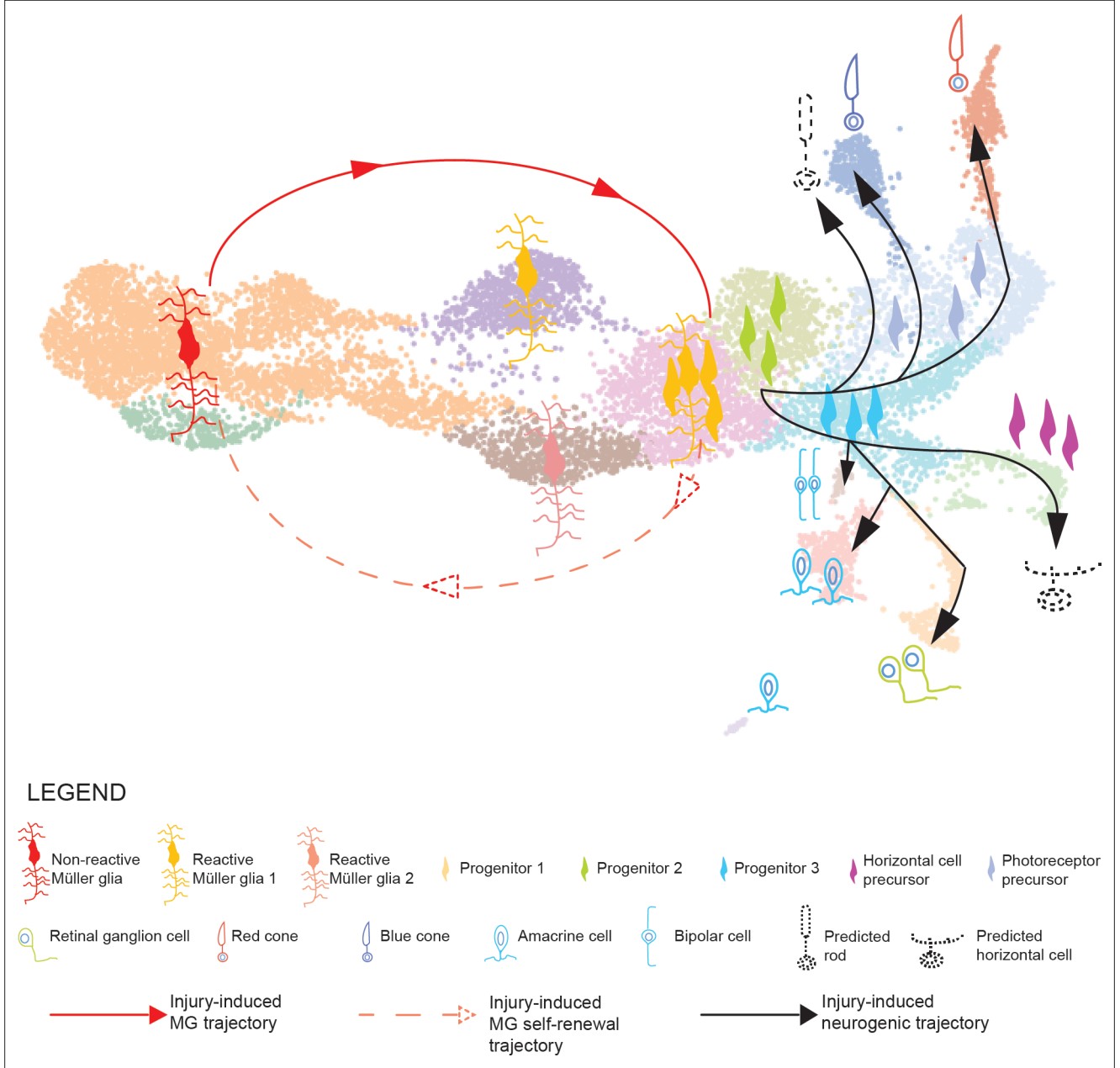

**Figure 10.** Model summarizing the cellular events underlying retina regeneration in zebrafish. Upon lesion, non-reactive MG on the left side of the main transcriptional UMAP feed into a hybrid Müller glia/progenitor population via reactive Müller glia (solid red line). Based from this hybrid cell population, we find a Müller glia trajectory reverting to the initial starting point indicating Müller glia self-renewal (dashed red line). Moreover, the hybrid cell population represents the tip of the neurogenic trajectory, resulting in the formation of neuronal precursors and differentiating neurons (solid black lines). Differentiating rods and horizontal cells, which have not been found in the scRNAseq, are indicated with dashed lines and positioned in the predicted location on the main UMAP.

a trajectory from non-reactive MG on the left through reactive MG in the middle, to photoreceptor precursors on the right side of the UMAP in the uninjured retina (*Figure 2B*). However, some cells end up in the amacrine as well as retinal ganglion cell clusters. This suggests that MG might have the potential to generate not only rods, but also all other retinal neurons in the homeostatic adult retina. Indeed, MG in the uninjured retina express, albeit at a low level, multipotency markers like *pax6*, and share many marker genes with RPCs (*Bernardos et al., 2007*; *Kim et al., 2005*; *Raymond et al., 2006*). Still, MG and injury-induced RPCs are considered distinct cell populations. Upon injury, MG divide only once between 31 and 48 hpl, whereas MG-derived RPCs continue to divide, reaching

a peak of proliferation at 4–5 dpl (discussed in *Fausett and Goldman, 2006*; *Lahne et al., 2020b*; *Lenkowski and Raymond, 2014*; *Thummel et al., 2008*). We isolated cells at 44 hpl, a time point when MG are about to undergo, or already underwent, their asymmetric cell division (*Bernardos et al., 2007*; *Lahne et al., 2015*; *Lenkowski and Raymond, 2014*; *Nagashima et al., 2013*) and identify cells in cluster 3 as early reactive MG. Consistent with that, we find that cells in cluster 3 do not only express *gfap* and *apoeb*, but also *pcna* (*Figures 2D and 3A*). Moreover, the predicted cell cycle status supports this notion. Whereas cells in the left part of cluster 3 are mostly in G1, cells in G2/M phase are found in the right part, with cells in S phase overlapping with both domains (*Figure 2C*). Cells in cluster 3 also upregulate inflammation-related markers like *crlf1a*, as well as *mmp9* and *hbegfa* (*Figures 3C and 4*, *Figure 3—figure supplement 1B* and *Supplementary file 1*). The upregulation of inflammatory genes makes MG similar to immune cells (*Silva et al., 2020*; *Wan et al., 2012*; *Zhao et al., 2014*). In this context, our population of early reactive MG (cluster 3), identified mostly at 44 hpl, resembles the reactive MG pool found in the regenerating zebrafish retina at 36 hpl (*Hoang et al., 2020*). However, our MG cluster is proliferative, whereas the one discussed by the mentioned study is not. The presence of a distinct pool of non-proliferative, reactive MG as early as 36 hpl in the earlier study might be explained by the different time points of sample collection, and/or by the lower number of cells sequenced in our experiment (11,690) in comparison to the previous one (45,153; *Hoang et al., 2020*). In summary, MG in the homeostatic zebrafish retina behave like multipotent RPCs. Upon injury, reactive MG express genes associated with inflammation, like *crlf1a*, and undergo asymmetric cell division. Daughter cells derived from this cell division most likely map to cluster 5, which is discussed below.

## A crossroad in the glial trajectory: hybrid cells with MG and RPC characteristics

The close similarity between zebrafish MG and MG-derived RPCs at the level of expressed markers as well as cellular and subcellular morphology has been described extensively in the literature (*Fausett and Goldman, 2006*; *Lahne et al., 2020a*; *Lenkowski and Raymond, 2014*). Thus, we suggest that cells in cluster 5, which displays a transcriptional hybrid identity between MG and RPCs, are the product of the asymmetric cell division from the early reactive MG (cluster 3). Cells in cluster 5 are still *gfap* positive, and share marker expression with early reactive MG, like *her4.1* and *crlf1a* (*Figure 3A and C*). However, cluster 5 is highly proliferative and becomes only prominent at 4 dpl, a time point when predominantly RPCs are proliferating (*Bernardos et al., 2007*; *Fausett and Goldman, 2006*; *Lenkowski and Raymond, 2014*; *Nagashima et al., 2013*). Consistently, *pcna* is highly expressed in cluster 5 and the predicted cell cycle status indicates that most cells are in S or G2/M phase (*Figure 2C and D*). Moreover, previous studies have found *her4.1* and *crlf1a* in MG-derived RPCs in the light- as well as stab-lesioned retina, where they control RPC proliferation at 4 dpl (*Wan et al., 2012*; *Wilson et al., 2016*; *Zhao et al., 2014*). We found that *her4.1* at 4 and 6 dpl locates to the inner nuclear layer as well as to the outer nuclear layer, where presumptive RPCs reside at this time point (*Figure 3D*; *Campbell et al., 2022*; *Wilson et al., 2016*). In addition, cells in cluster 5 upregulate *sox9b* and *nr2e1*, which have been found in early, multipotent RPCs of the developing fish, avian and mammalian retinae, as well as in late retinogenesis, where they regulate MG genesis (*Buenaventura et al., 2018*; *Corso-Díaz and Simpson, 2015*; *Kitambi and Hauptmann, 2007*; *Muto et al., 2009*; *Poché et al., 2008*). Hence, we propose that cells in cluster 5 represent a crossroad of the glial and neurogenic trajectory, containing self-renewed MG as well as first-born, MG-derived RPCs as a direct result of the earlier (44 hpl) asymmetric MG cell division. We further propose that self-renewed MG close the cycle via late reactive MG (cluster 4), which is evident at 4 dpl, and return into non-reactive MG (clusters 1 and 2, red, dashed line in *Figure 10*). In this respect, we note that reactive MG 2 (cluster 4) share many marker genes with early reactive MG 1 in cluster 3 (*Figure 2—figure supplement 2*, *Supplementary files 1 and 3*). Both reactive MG cell populations express marker genes that are involved in inflammation, like *crlf1a* (*Figure 3C*, *Supplementary file 1*) encoding for an inflammatory cytokine (*Zhao et al., 2014*). However, proliferation markers like *pcna* are expressed only in reactive MG in cluster 3 but not in cluster 4 (*Figure 3—figure supplement 1*). We corroborated *crlf1a* expression in the regenerating tissue using fluorescent in situ hybridization combined with immunohistochemistry for PCNA and GFAP (*Figure 4*). The staining confirmed co-localization of the transcript with PCNA-, GFAP-double positive cells at 44 hpl as well as at 4 dpl. The co-localization of *crlf1a* with PCNA at 4

dpl does not seem to match the sequencing data for reactive MG 2, which is eminent at 4 dpl and should not be proliferative. However, it is possible that *crlf1a*/PCNA-double positive cells detected at 4 dpl are still positive for the PCNA protein, but no longer express the *pcna* transcript. Double in situ hybridization for *pcna* and *crlf1a* would be needed to fully address whether *crlf1a*-positive cells are still *pcna*-positive at 4 dpl. It is also possible that *crlf1a*/GFAP-double positive but PCNA-negative Müller glia are fewer and only masked in the crowd of *crlf1a*-, PCNA-double positive, GFAP-negative progenitors at 4 dpl (*Raymond et al., 2006*). Nevertheless, we would like to mention that we cannot exclude the possibility that non-proliferative, reactive MG in cluster 4 represent MG that are reactive to the tissue injury, but never re-enter the cell cycle upon light-lesion.

In contrast to MG, MG-derived RPCs embark into the neurogenic pathway (cluster 6, *Figure 10*, solid, black line). In an alternative model, cells in cluster 5 might represent only MG cells that have upregulated early RPC markers like *sox9b* and *nr2e1* and are dividing a second time at 4 dpl to generate renewed MG (cluster 4) and neurogenic RPCs (cluster 6). However, MG have been described to divide only once between 31 and 48 hpl and additional and/or asynchronous MG divisions later than 2 dpl have not been investigated (*Lahne et al., 2015*; *Lenkowski and Raymond, 2014*; *Nagashima et al., 2013*; *Thummel et al., 2008*). BrdU/EdU lineage tracing experiments will be required to address this issue, with the caveat that MG and RPCs are hardly distinguishable from each other in the 4 dpl retina, especially when using the Zrf1 antibody to label MG (*Fausett and Goldman, 2006*; *Raymond et al., 2006*). Live imaging of the regenerating retina would be the best tool to follow MG and MG-derived progenitors in vivo, but the technique is not optimized yet for long recordings from the intact tissue in the living animal (*Lahne et al., 2015*). Either way, our proposed model of self-renewing MG is further supported by the change of cell density over time (*Figure 2B*). Following the cells across the four time points on the UMAP, we observe that initially most cells of the uninjured sample map to non-reactive MG (clusters 1 and 2) located at the far left. At 44 hpl, the cell density shifts towards the upper right portion of cluster 1 and towards the early reactive MG (cluster 3) in the upper, middle of the map. At 4 dpl, sampled cells mostly map to RPCs (clusters 5–9) on the right, as well as to late reactive MG (cluster 4) in the lower, middle part. Additionally, cells mapping to the lower right part of non-reactive MG (clusters 1 and 2) are enriched in this sample, indicating a reverse direction from right to left. This population is significantly further increased at 6 dpl, when the number of RPCs is less prominent and the differentiating neural types at the top and bottom right are predominantly present in the sample. According to this model, sample collection at even later time points, for example 10 or 14 dpl, might show that the initial status has been restored and that non-reactive MG are mostly enriched. However, it is unlikely that this experiment could be performed with our short-term labelling strategy.

## The neurogenic trajectory produces regenerated progeny that partially follow the developmental order of neurogenesis

Neurogenic RPCs map to the right of cluster 5, and include proliferating, neurogenic RPCs (cluster 6) and the downstream, non-proliferating, neurogenic RPCs (cluster 7). Cells in these clusters express *insm1a*, *atoh7* and additional neurogenic genes that are required for the generation of several neuronal lineages (*Figure 7*, *Figure 3—figure supplement 1D* and *Supplementary files 1 and 6*). Indeed, *atoh7* expression has been described to be initiated in RPCs on their way to the last cell division giving rise to a retinal ganglion cell and another cell that can be either an amacrine cell, or a precursor of a horizontal or of a cone cell in the developing zebrafish retina (*Amini et al., 2017*; *Nerli et al., 2022*; *Poggi et al., 2005*). Cluster 6 and 7 appear also highly similar to the transient, neurogenic RPC state that has been described recently in the developing mouse retina (*Wu et al., 2021*). We propose that cluster 6 and 7 are the roots of the neurogenic trajectory from which fate-restricted RPCs and regenerated progeny arise at 4 and 6 dpl (solid, black line in *Figure 10*). Cells in cluster 6 are *mki67*-positive, but additionally express *insm1a*, which promotes RPC cell cycle exit at 4 dpl in the injured retina (*Ramachandran et al., 2012*). Cells in cluster 6 might undergo their last cell division, as suggested by the concomitant expression of *atoh7*. Consistently, in our scRNAseq high counts of *atoh7* are found in cells of cluster 6, to cluster 7 and eventually cluster 10. Cluster 10 represents differentiating retinal ganglion cells that are the first regenerated progeny, evident as early as 4 dpl (*Figures 7 and 10* and *Figure 8—figure supplement 1*).

Clusters 14 and 15 highly express amacrine cell genes (*Figure 8—figure supplement 2B*). Interestingly, our fluorescent-reporter sorting strategy allows the discrimination of distinct cell subsets

belonging to the same cell type. In this respect, cluster 14 includes GABAergic amacrine cells, as shown by the expression of *gad1b*, which encodes for the GABA-synthesizing enzyme GAD67 (*Bosma et al., 1999*). In contrast, cluster 15 includes a class of amacrine cells known as starburst amacrine cells, as indicated by the expression of both *gad1b* and *chata* (*O'Malley et al., 1992*). While amacrine cells arise directly from the core of neurogenic RPCs, fate restricted precursors for regenerated cones and for horizontal cells are identified (*Figures 8 and 10*), as it occurs during development (*Godinho et al., 2007*; *Weber et al., 2014*). Consistently with the events during development, red cones (cluster 11) branch from *thrb*-positive cells of the photoreceptor RPC cluster (cluster 8). Interestingly, *thrb* transcripts are found in neurogenic RPCs of cluster 7 as well as at the beginning of cluster 9, which contains horizontal cell precursors becoming evident at 4 dpl (*Figure 8B*). This is in agreement with observations in the developing zebrafish retina, where horizontal and cone cells share a common Thrb-positive progenitor (*Suzuki et al., 2013*). Mature markers of horizontal cells, like calbindins, are not found in cluster 9 (*Figure 8B*, *Figure 8—figure supplement 2C* and *Supplementary file 1*), in contrast to cone photoreceptor mature markers, like red light- and blue light-sensitive opsins, at 4 and 6 dpl. We did not detect UV-sensitive opsin marker gene expression labelling UV cones, which might arise later than 6 dpl and were hence not included in our sorted samples. While it might be that mature horizontal cells arise later than 6 dpl, their absence from our dataset could also be due to our experimental set-up. As explained before, we isolated mCherry-only positive and EGFP/mCherry-double positive cells to obtain MG and MG-derived cells. Whereas mCherry, driven by a *gfap* promoter fragment, is actively transcribed in MG, the label of MG-derived cells is completely dependent on the half-life of the fluorophore. Protein turnover and further cell divisions would result in a constant depletion of mCherry in MG-derived cells over time. In this context, we find a strong expression of *pcna* in cluster 9, indicating that additional cell divisions might have diluted the mCherry label to undetectable levels (*Figure 2D*). Inclusion of EGFP-positive only cells in our sample collection may reveal the pool of mature horizontal cells, which inherit the fluorophore from their *pcna:EGFP*-positive precursors. Accordingly, we find the presence of horizontal cell precursor markers in our dataset. For instance, cells in cluster 9 express *onecut1* and *onecut2*, which encode transcription factors and are upregulated in horizontal cell RPCs in the developing mouse retina, and which regulate horizontal cell maintenance in adult animals (*Emerson et al., 2013*; *Klimova et al., 2015*; *Sapkota et al., 2014*). Counts for *onecut1* and *onecut2* are found also in earlier neurogenic RPCs of clusters 6 and 7, where they most likely contribute to the transient state of RPCs together with additional neurogenic markers, like *atoh7* (*Wu et al., 2021*). This is again consistent with observations in the developing mouse retina, where Onecut1 and Onecut2 regulate the development of retinal ganglion cells, as well as the development of cones, horizontal cells, and of a subset of amacrine cells, while inhibiting rod specification genes (*Emerson et al., 2013*; *Sapkota et al., 2014*).

In addition to mature horizontal cells, we were also unable to detect any rod-specific cluster expressing mature rod markers like *rho* and *gnat1* (*Figure 8—figure supplement 2A*). We assume that mature rods might arise as a third branch from the photoreceptor RPC cluster (*Figure 10*), which indeed expresses rod specification markers like *nr2e3* (*Supplementary file 1*). We speculate that rods either become evident only later than 6 dpl, or that they do not survive the isolation procedure, since rods possess a complex morphology with long outer segments. A third possibility might be that rod photoreceptors are replenished by their dedicated rod precursors faster than the triggering of the MG-dependent injury response. In this context, rod precursors form a pool of cells at the base of the outer nuclear layer and proliferate as early as 16 hours post-light lesion (*Morris et al., 2005*; *Stenkamp, 2011*; *Thummel et al., 2008*). Since the already available pool of rod precursors would be *gfap*:mCherry-negative, we might have missed the derived rod progeny, given our sorting strategy. In summary, we show that in the light-lesioned retina, zebrafish MG go through an inflammatory-like state before generating hybrid MG/RPCs that eventually cycle-back to non-reactive MG as well generate neurogenic RPCs. The neurogenic RPCs progressively restrict their competence to produce neurons that include retinal ganglion cells first and bipolar cell last up to 6 dpl. A previously published study used scRNAseq to profile the transcriptome of cells in the zebrafish retina upon several lesion paradigms, including the light injury, up to 72 hpl (*Hoang et al., 2020*). In their study, the authors dissociated uninjured and lesioned retinae to single cells that were all subsequently subjected to scRNAseq. Conversely, in our study, we used a short-term lineage tracing strategy to enrich for and sequence specifically MG and MG-derived cells in uninjured and light-lesioned retinae up to 6 dpl.

Bioinformatic comparison between the two datasets revealed a high degree of overlap between the respective cell clusters, as it would be expected when the same tissue is analysed. Specifically, Hoang and colleagues annotated resting MG and activated MG, as well as progenitor clusters that all map to non-reactive MG, reactive MG and progenitor clusters in our scRNAseq dataset, respectively (*Figure 9* and *Figure 9—figure supplement 1*). In contrast to Hoang and colleagues, we were able to capture a certain degree of heterogeneity in MG and progenitor populations, and thus identify two clusters of non-reactive MG, two clusters of reactive MG and five clusters of progenitors and neuronal committed precursors. Consistently with the heterogeneity of MG described in our work, a recent study found MG heterogeneity in both quiescent and lesioned zebrafish retina using scRNAseq as well (*Krylov et al., 2023*). Specifically, this study reports six clusters of quiescent Müller glia that display differential spatial distribution along the dorsal/ventral retinal axis. The authors describe a ventral, quiescent Müller glia population that shares some marker genes (*aldh1a3*, *rdh10a*, *smoc1*) with our non-reactive Müller glia 2 (cluster 2, *Supplementary files 1 and 2*). Furthermore, *Krylov et al., 2023* show that Müller glia that locate to different spatial position show differential activation and proliferation response upon photoreceptor ablation (*Krylov et al., 2023*). Due to the half-life of mCherry and EGFP, we were also able to sample MG-derived cells to capture large parts of the regenerated neurons and birthdate their order specifically. This is in contrast to Hoang and colleagues, who on the one hand sequenced all retinal cells, independently of their contribution to regeneration, and only used samples up to 72 hpl, thus lacking sufficient MG lineage resolution to detect lesion-specific, regenerated progeny (*Hoang et al., 2020*). While they were able to annotate clusters for almost all mature retinal cell types, the applied scRNAseq strategy is unable to distinguish between neurons that were already present in the retina and those that were regenerated specifically upon lesion. Given that we only sequenced MG and MG-derived cells, this could explain why Hoang and colleagues detected mature horizontal and rod cells, while we could not. Interestingly, and despite this discrepancy, horizontal cells detected by Hoang and colleagues share some marker genes with the horizontal cell precursors we annotated in our dataset (*Figure 9—figure supplement 1*). Likewise, rod photoreceptors from the Hoang dataset map largely, although not exclusively, to photoreceptor precursors in our dataset (*Figure 9* and *Figure 9—figure supplement 1*). Importantly, there is a consistent overlap between retinal ganglion, cone, amacrine and bipolar cells from Hoang and colleagues' dataset and the corresponding cell clusters in our dataset, which further strengthens our results.

In summary, our study is the first comprehensive description of the molecular network underlying MG self-renewal and RPC differentiation to regenerating neurons which will inform future regenerative therapies to treat retinal dystrophies in humans.(*Javed and Cayouette, 2017*).

# Materials and methods

## Key resources table

| Reagent type (species) or resource | Designation | Source or reference | Identifiers | Additional information |
|---|---|---|---|---|
| Antibody | Zrf1 (Mouse, monoclonal, IgG1) | ZIRC | RRID:AB_10013806 | IHC, 1:250 |
| Antibody | Anti-mCherry (Rabbit, polyclonal) | Takara Bio | Catalog #632475; RRID:AB_2737298 | IHC, 1:500 |
| Antibody | Anti-PCNA (Mouse, monoclonal, IgG2a) | Dako | RRID:AB_2160651 | IHC, 1:500 |
| Antibody | Anti-GFP (Chicken, polyclonal) | Abcam | RRID:AB_300798 | IHC, 1:2000 |
| Antibody | Sheep anti-Digoxigenin-AP Fab fragments | Roche | Catalog #11093274910; RRID: AB_514497 | ISH, 1:2000 |
| Antibody | Sheep anti- Digoxigenin-POD Fab fragments | Roche | Catalog #11207733910; RRID:AB_514500 | ISH, 1:2000 |
| Antibody | Alexa Fluor 633 Goat anti-Mouse IgG1 | Thermo Fisher Scientific | Catalog #A21126; RRID: AB_2535768 | IHC, 1:500 |

*Continued on next page*

*Continued*

| Reagent type (species) or resource | Designation | Source or reference | Identifiers | Additional information |
|---|---|---|---|---|
| Antibody | Alexa Fluor 555 Goat anti-Rabbit | Thermo Fisher Scientific | Catalog #A21429; RRID: AB_2535850 | IHC, 1:500 |
| Antibody | Alexa Fluor 488 Goat anti-mouse IgG2a | Thermo Fisher Scientific | Catalog #A21131; RRID: AB_2535771 | IHC, 1:500 |
| Chemical compound, drug | NBT/BCIP stock solution | Roche | Catalog #11681451001 | ISH, 20 µl/ml |
| Chemical compound, drug | Blocking reagent | Roche | Catalog # 11096176001 | ISH |
| Chemical compound, drug | Calcein blue, AM | Invitrogen | Catalog #C1429 | Living cell dye, 1 µM |
| Chemical compound, drug | Neutral protease (Dispase) | Worthington Biochemical Corporation | Catalog #LS02100 | 0.2 U/ml |
| Commercial assay or kit | Papain dissociation System | Worthington Biochemical Corporation | Catalog #LK003150 | |
| Commercial assay or kit | 10 x Chromium single - cell kit | 10 x Genomics | | V3 Chemistry |
| Commercial assay or kit | Total RNA purification kit | Norgen Biotek Coporation | Catalog #17200 | |
| Commercial assay or kit | cDNA synthesis | Roche | Catalog #4379012001 | |
| Commercial assay or kit | TOPO TA cloning dual promoter | Thermo Fisher Scientific | Catalog #450640 | |
| Commercial assay or kit | Zymoclean Gel DNA Recovery Kit | Zymo Research | Catalog #D4002 | |
| Commercial assay or kit | Gene Jet Plasmid Miniprep kit | Thermo Fisher Scientific | Catalog #k0503 | |
| Commercial assay or kit | Nucleo Bond Xtra Midi | Macherey-Nagel | Catalog #740410.5 | |
| Commercial assay or kit | TSA Plus Cyanine 3 System, | AKOYA Biosciences | Catalog #NEL744A001KT | |
| Other | 20 x SSC buffer | | | 175.3 g NaCl, 88.2 g HOC(COONa)(CH2COONa)2 · 2H2O, adjust pH to 6 with acetic acid in DEPC water |
| Other | Washing buffer | | | 50% deionized formamide, 1 x SSC in DEPC water |
| Other | MABT buffer | | | 100 mM maleic acid, 150 mM NaCl, adjust pH to 7.5 with NaOH, 0.1% Tween 20, in DEPC water sterile filter |
| Other | Staining buffer for Alkaline Phosphatase | | | 100 mM NaCl, 50 mM MgCl2, 100 mM Tris base, pH = 9.5, 0.1% Tween 20 in DEPC water |
| Other | TNT buffer | | | 100 mM Tris HCl, pH = 7.5, 150 mM NaCl, 0.1% Tween 20 in DEPC water |

## Animals

Zebrafish (*Danio rerio*) were raised as previously described (*Brand et al., 2002*). Adult fish (6–12 months old) of either sex were used for all experiments. WT animals were in the AB genetic background.

## Transgenic lines

To generate the *Tg(pcna:EGFP)* (tud119), the BAC CH73-140L11, containing more than 100 kb of the genomic *pcna* locus, was obtained from the BACPAC resource centre. A cassette containing EGFP and a polyadenylation signal was recombined at the starting ATG of exon 1. First, FRTgb- neo-FRT and EGFP cassettes where amplified by PCR. Then, the cassettes where amplified and merged by fusion PCR with primers carrying 50 nucleotides of homology of the targeting sequence. Recombineering of the BAC was performed as previously described (*Zhang et al., 1998*). BAC DNA was purified and about 50 pg of BAC DNA were injected into fertilized eggs at the one-cell stage. F0 were raised and incrossed. F1 were identified by visual screening for GFP expression and raised. The *Tg(gfap:nls-mCherry)* line was generated as previously described (*Lange et al., 2020*). In addition, to the line described in Lange et al. (tud117tg), a second independent insertion was isolated (tud120). This insertion displays a stronger expression in MG and was hence used for this study.

## Light injury

WT AB and *Tg(pcna:EGFP);Tg(gfap:nls-mCherry)* fish were dark adapted for 3 days. Afterwards, diffuse light lesion was performed as previously described (*Weber et al., 2013*). Briefly, up to 4 fish were placed in a beaker filled with system water (300 ml), positioned 3 cm away from the light bulb of a metal halide lamp (EXFO X-Cite 120 W, EXFO Photonic Solutions, Mississauga, Ontario, Canada). Fish were exposed for 30 min to a bright light (≥100,000 lux), and then put back to system tanks in normal light conditions. Uninjured fish controls were dark adapted and processed in parallel to light injured animals.

## Dissociation of retinae to a single-cell suspension

Uninjured and light-lesioned *Tg(pcna:EGFP);Tg(gfap:nls-mCherry)* fish were exposed to an overdose (0.2%) of MS-222 (Sigma) (*Brand et al., 2002*) until cessation of the operculum movement. Subsequently, each cornea was poked with a needle, and tweezers were gently pressured on each side of the eye to remove the lens. Then, tweezers were used to carefully isolate the retinae, which were put in a dissociation solution of papain (16 U/ml), dispase (0.2 U/ml) and DNAse (168 U/ml) dissolved in Earle's Balanced Salt Solution (EBSS, Worthington Biochemical Corporation). Up to 6 retinae were put in 1.7 ml of dissociation solution. Retinae were digested for 30 min at 28 °C, and then calcein blue (final concentration: 1 µM, Invitrogen) was added for 10 min to the digestion solution at 28 °C to stain for living cells. At the end of the enzymatic digestion, mechanical trituration was applied using fire-polished glass pipettes of decreasing tip diameter to help tissue dissociation. Single cells were centrifuged (300 *g*) at 4 °C, for 6 min. The cell pellet was re-suspended in EBSS supplemented with DNAse (100 U/ml) to eliminate residual, ambient DNA, and albumin ovomucoid inhibitor (1:10 of stock solution, Worthington Biochemical Corporation) to stop papain action. The singlecell solution was eventually filtered (20 µm filters) to remove clumps and debris, and put on ice, ready for Fluorescent Activated Cell Sorting (FACS). Each single-cell suspension was obtained by pooling 8 retinae (from 4 fish, 2 males and 2 females) of uninjured and light-lesioned *Tg(pcna:EGFP);Tg(gfap:mCherry)* animals. Uninjured WT AB, *Tg(gfap:mCherry)* and *Tg(pcna:EGFP)* fish were used to prepare single cell suspensions of dissected retinae for setting the single fluorophore gates in flow cytometry. Additionally, a calcein only stained single cell suspension was prepared from uninjured WT AB fish retinae for the setting of the calcein gate in the flow cytometry.

## Flow cytometry and FACS

A BD FACSAria III cell sorter was used for flow cytometry and FACS. mCherry fluorophore was detected using a 561 nm excitation laser and a 610/20 nm band pass filter, EGFP using a 488 nm laser and a 530/30 nm band pass filter and calcein blue using a 405 nm laser and a 450/40 nm band pass filter. The gating strategy is shown in *Figure 1—figure supplement 3* for uninjured samples and was the same for all the light-lesioned samples. Forward and side scatters were used to gate cells among all the events in the single cell suspension and to gate singlets among those cells (*Figure 1—figure supplement 3A*). mCherry-positive only as well as EGFP-positive only gates were set in comparison to the unstained, WT control (*Figure 1—figure supplement 3B*). Calcein blue positive, single cells were eventually gated as living cells (60,000 living cells from uninjured, 44 hpl and 4 dpl samples, 41,389

living cells from 6 dpl sample). 15,000 cells were sorted from each samples, put on ice and sent to the CMCB Deep Sequencing Facility for single cell RNA sequencing (scRNAseq).

## (Droplet based) single-cell transcriptomics

Single cell transcriptome sequencing was performed based on the 10 x Genomics Single-cell transcriptome workflow (*Zheng et al., 2017*). Specifically, 15,000 FACS-sorted cells were recovered in BSA-coated tubes containing 5 ul PBS with 0.04% BSA. Cell samples were carefully mixed with Reverse Transcription Reagent (Chromium Single Cell 3' Library Kit v3) and loaded onto a Chromium Single Cell B Chip to reach a recovery of 10,000 cells per sample. The samples were processed further following the guidelines of the 10 x Genomics user manual for single cell 3' RNA-seq v3. In short, the droplets were directly subjected to reverse transcription, the emulsion was broken and cDNA was purified using silane beads (Chromium Single Cell 3' Gel Bead Kit v2). After amplification of cDNA with 11 cycles, samples were purified with 0.6 x volume of SPRI select beads to deplete DNA fragments smaller than 400 base pairs and cDNA quality was monitored using the Agilent FragmentAnalyzer 5200 (NGS Fragment Kit). Ten µl of the resulting cDNA were used to prepare single cell RNA seq libraries - involving fragmentation, dA-Tailing, adapter ligation and 11 cycles of indexing PCR following manufacturer's guidelines. After quantification, both libraries were sequenced on an Illumina NextSeq500 in 75 base pair paired-end mode, generating 45–109 million fragments per transcriptome library. The raw sequencing data was processed with the 'count' command of the Cell Ranger software (v2.1.0) provided by 10 X Genomics. To build the reference, the zebrafish genome (GRCz11) as well as gene annotation (Ensembl 90) were downloaded from Ensembl and the annotation was filtered with the 'mkgtf' command of Cell Ranger (options: '—e attribute = gene_biotype:protein_coding --attribute=gene_biotype:lincRNA –attribute = gene_biotype:antisense'). Genome sequence and filtered annotation were then used as input to the 'mkref' command of Cell Ranger to build the appropriate Cell Ranger.

## Bioinformatic analysis

The complete software stack for the analysis is available as a Singularity container (https://gitlab.hrz.tu-chemnitz.de/dcgc-bfx/singularity/singularity-single-cell, tag 1f5da0b; *Petzold, 2023*). The count matrices generated with Cell Ranger were further analysed using scanpy 1.8.2 (*Wolf et al., 2018*). For quality control, we removed cells with less than 2.000 counts, less than 500 detected genes, more than 15% mitochondrial reads or more than 60% of the counts in the top 50 highest expressed genes. The counts were normalised with the size factors computed with calculateSumFactors from scran 1.18.5 (min.mean=0.1)(*Lun et al., 2016*). The normalised counts were log-transformed using the numpy function log1p. The top 4000 highly variable genes were detected using the scanpy function scanpy.pp.highly_variable_genes (n_top_genes = 4000). Cell cycle scores were computed using scanpy.tl.score_genes_cell_cycle on each sample separately. As genes associated to S-phase, the genes *mcm5, pcna, tyms, fen1, mcm2, mcm4, rrm1, unga, gins2, mcm6, cdca7a, dtl, prim1, uhrf1, si:dkey-185e18.7, hells, rfc2, rpa2, nasp, rad51ap1, gmnn, wdr76, slbp, ccne2, ubr7, pold3, msh2, atad2, rad51, rrm2, cdc45, cdc6, exo1, tipin, dscc1, blm, casp8ap2, usp1, pola1, chaf1b, brip1* and *e2f8* were used. As genes associated to G2/M phase, the genes *hmgb2a, cdk1, nusap1, ube2c, birc5a, tpx2, top2a, ndc80, cks2, nuf2, cks1b, mki67, tmpoa, cenpf, tacc3, smc4, ccnb2, ckap2l, aurkb, bub1, kif11, anp32e, tubb4b, gtse1, kif20ba, si:ch211-69g19.2, jpt1a, cdc20, ttk, kif2c, rangap1a, ncapd2, dlgap5, si:ch211-244o22.2, cdca8, ect2, kif23, hmmr, aurka, anln, lbr, ckap5, cenpe, ctcf, nek2, g2e3, gas2l3, cbx5* and *selenoh* were used. A principal component analysis (PCA) was performed on the highly variable genes (function scanpy.pp.pca using svd_solver='arpack'). A k-nearest-neighbour (knn) graph was computed using scanpy.pp.neighbors and a UMAP was computed using scanpy.tl.umap. Clustering was performed with scnapy.tl.leiden (resolution = 0.01). For the three resulting clusters, marker genes were computed using scanpy.tl.rank_genes_groups. Two smaller clusters identified as microglia and low-quality cells were removed from the downstream analysis. Detection of highly variable genes PCA were performed as above on the remaining cells. A new knn graph was computed (this time with n_neighbors = 50) and a new UMAP was computed. Embedding densities were computed using scanpy.tl.embedding_density. Clustering was performed in multiple steps, always using scanpy.tl.leiden: First, the data were clustered with the resolution parameter set to 0.45. Next, cluster 10 was subclustered with resolution 0.1 and cluster 0 was subclustered with resolution 0.2. Marker genes of

the resulting 15 clusters were computed with scanpy.tl.rank_genes_groups. Differential expression analysis of pairs of clusters was performed by subsetting the data to cells from those two clusters and computing differentially expressed genes with scanpy.tl.rank_genes_groups. The source code for the analysis is available on GitHub (https://github.com/fbnrst/celotto_2023).

### Cluster annotation

Cluster annotation was performed in several steps. Initially, we identified clusters by considering the top 100 upregulated genes per cluster, the time point at which they appeared and their cell cycle profile. Genes were manually annotated by consulting the ZFIN database (http://zfin.org/) and Pubmed (https://pubmed.ncbi.nlm.nih.gov/) indexed literature about retina development and regeneration. In particular, we took as a reference the single cell RNA sequencing dataset of the regenerating zebrafish retina from *Hoang et al., 2020*, for marker genes lists. Following the initial identification of cell clusters, we proceeded with a finer characterization by considering differential gene expression analysis for clusters that shared a similar identity. Occasionally (e.g. in the case of horizontal cell precursor marker *ptf1a*), we plotted the UMAP profile of known candidates from retina development or regeneration. Selected marker genes were then checked for expression in the retina tissue via in situ hybridization, fluorescent in situ hybridization and immunohistochemistry on cryosections.

### Comparison with Light Damage data in Hoang et al., 2020

The count matrix, for the light damage single-cell data was downloaded from https://github.com/jiewwwang/Single-cell-retinal-regeneration ( *Wang, 2022*; *Hoang et al., 2020*). The data was used as a reference data for Scanpy's ingestion method (*Wolf et al., 2018*). The light-damage data was normalized with scanpy's pp.normalize_total and then log-transformed pp.log1p. This log-normalised count-matrix was used to create the dotplot in *Figure 9—figure supplement 1*. Next, our data and the light-damage data were subsetted to the set of genes that was present for both count matrices. A total of 3000 highly variable genes were computed using pp.highly_variable_genes, and then 50 principal components were computed using pp.pca. Then, the cell types from the light-damage reference data were ingested onto our data using tl.ingest.

### Tissue preparation and cutting

Fish heads were fixed with 4% paraformaldehyde in 0.1 M phosphate buffer (PB, pH: 7.4) overnight, at 4 °C. Subsequently, fixed heads where decalcified and cryoprotected in 20% EDTA and 20% sucrose in 0.1 M PB (pH: 7.5) overnight, at 4 °C. Heads were eventually embedded in 7.5% gelatin, 20% sucrose in 0.1 M PB and sectioned (12 or 14 μm) at the cryostat (CryoStar Nx70, Thermo Scientific). Cryosections were stored at –20 °C.

### in situ hybridization probe synthesis

For the synthesis of in situ hybridization RNA probes, 4 dpl retinae were dissected and sonicated at 4 °C to homogenate the tissue and to extract total RNA. We chose 4 dpl samples, because transcripts of interest were enriched at 4 dpl. RNA was extracted using the total RNA purification kit and following manufacturer's instructions (Norgen Biotek Corporation). cDNA was then transcribed using the first strand cDNA synthesis kit (Roche), followed by PCR to amplify the fragments of interest using primer pairs listed in the key resource table. The amplicon was subsequently purified (gel and DNA recovery kit, Zymo Research) and cloned in a TOPO TA vector (Thermo Fisher), with which competent *E coli* bacteria were transformed and cultured on Petri dishes coated with agar and ampicillin, overnight at 37 °C. The next day, colonies were picked up and cultured in LB medium supplemented with ampicillin (1:2000) overnight at 37 °C, shaking, to enrich cultures for mini-prep (Gene Jet Plasmid Miniprep kit k0503, Thermo Fisher). Purified plasmids were sent for sequencing (Eurofins) to verify the sequence. Insert-positive colonies were further enriched in LB/ampicillin cultures for midi-prep (Nucleo Bond Xtra Midi, Macherey – Nagel) and purified plasmids were linearized to transcribe antisense, digoxigenin labelled (DIG) RNA probes, which were stored in hybridization buffer at –20 °C. Probes for *her4.1* (*Takke et al., 1999*), *stm* (*Söllner et al., 2003*), *crlf1a, hmgb2b, atoh7, onecut1* (*Matthews et al., 2004*), *onecut2* were transcribed from a TOPO TA cloned insert. Instead, the *id1* probe was transcribed directly from the PCR – amplified cDNA using SP6-conjugated forward and T7-conjugated reverse primers (key resource table).

## in situ hybridization

in situ hybridization and fluorescent in situ hybridization were performed on three animals per injured time point and on three uninjured controls processed in parallel. Cryosections were dried for 30 min at room temperature, then washed two times for 10 min in phosphate buffer saline supplemented with 0.3% Triton X-100 (PBSTx) for permeabilization. in situ hybridization probes were denatured by warming up at 70 °C for 10 min and brief vortexing. Permeabilized sections were incubated overnight in denatured probes at 62 °C. The next day, sections were rinsed three times (once for 15 min, followed by two washes of 30 min each) in washing buffer (refer to key resource table) at 62 °C. Then, in the case of chromogenic in situ hybridization, sections were washed twice in MABT (key resource table) for 30 min each and incubated with 2% digoxigenin blocking reagent (key resource table) in MABT for 1 hr at room temperature. In the case of fluorescent in situ hybridization, sections were washed in TNT buffer (key resource table), and then incubated with 2% digoxigenin blocking reagent dissolved in TNT buffer. After, sections were incubated with the AP-conjugated, anti-digoxigenin antibody (chromogenic in situ hybridization, 1:2000, key resource table) or with the POD-conjugated, anti-digoxigenin antibody (fluorescent in situ hybridization, 1:2000, key resource table) overnight, at 4 °C. The next day, sections were washed four times for 20 min each in MABT (chromogenic in situ hybridization) or three times for 10 min each in TNT buffer (fluorescent in situ hybridization) at room temperature. Then, sections for chromogenic in situ hybridization were washed for 5 min in staining buffer (key resource table) and eventually incubated with 20 µl/ml NBT/BCIP stock solution in staining buffer, at room temperature, to develop the colour reaction. Time of incubation varied between 3 hr and overnight, depending on the target mRNA and probe concentration. In the case of fluorescent in situ hybridization, after TNT washing steps, sections were incubated for 30 min with 1:200 Cy3-Tyramide in TSA amplification buffer (Akoya Biosciences), at room temperature. Finally, sections from chromogenic in situ hybridization were washed three times with PBS, for 10 min each and mounted in 70% glycerol in PBS, ready for bright field microscopy. Fluorescent in situ hybridization sections were rinsed three times in PBS, for 10 min each, and then incubation with primary antibody for immunohistochemistry was performed as described below.

## Immunohistochemistry

Immunohistochemistry was conducted as previously described (*Grandel et al., 2006*; *Kroehne et al., 2011*). Immunohistochemistry was performed on three light-lesioned animals and three, parallel processed, uninjured control per condition. Sections were dried at room temperature for 30 min and washed three times in PBSTx for 10 min each. Then, primary antibodies diluted in PBSTx were applied overnight at 4 °C (refer to the key resource table for primary antibody host, antigen specificity and dilution). For proliferating cell nuclear antigen (PCNA) retrieval, sections were immersed in 10 mM sodium citrate (pH: 6) for 6 min at a temperature higher than 80 °C, then washed, and finally primary anti-PCNA antibody was applied. After primary antibody incubation, sections were washed three times in PBSTx for 10 min each, then Alexa fluorophore conjugated, secondary antibodies diluted In PBSTx were applied accordingly (refer to key resource table for secondary antibody host, primary antibody specificity and dilution) for 2 hr at room temperature. Sections were eventually washed three times in PBS, mounted in 70% glycerol in PBS, and stored at 4 °C. Each staining was performed on three fish per time point upon injury and on three uninjured fish controls processed in parallel. When combined with fluorescent in situ hybridization, immunohistochemistry followed fluorescent in situ hybridization processed sections.

## Microscopy

Bright field images of chromogenic in situ hybridizations and fluorescent images in Fiure1-figure supplement 2 were acquired with the Widefield ApoTome microscope (Zeiss Axio Imager.Z1) using the 20 x/0.8 Plan-Apochromat, Air, DIC objective. Fluorescent images in *Figure 1—figure supplement 1*, *Figure 3—figure supplement 2* and in *Figures 4 and 6* were acquired with the laser scanning confocal microscope 780 (Zeiss Axio Imager.Z1) using the 40 x/1.2 C-Apochromat, Water, DIC, Zeiss (*Figure 1—figure supplement 1* and *Figure 4*) and the 20 x/0.8 Plan-Apochromat, Air, DIC, Zeiss (*Figure 6*). Fluorescent channels were acquired sequentially to minimize crosstalk in multicolour specimens.

## Image acquisition and processing

Bright field microscope images were processed with the Zeiss ZEN blue, V 2012 software. The extended depth of focus function (contrast method) was applied to raw, bright field images of *her4.1*, *hmgb2b*, *atoh7* and *onecut1* in situ hybridizations to project the chromogenic signal of the optical stack to a single plane. Fluorescent microscope images were processed using Fiji (*Schindelin et al., 2012*) for image contrast adjustments, crop and rotation. Adobe Illustrator CC 2015.3 was used to assemble figures. Excel 2016 (Microsoft) was used to generate the plot in *Figure 8—figure supplement 1*.

## Statistics

Marker genes were ranked by their test statistic computed with a t-test for each gene, comparing cells within the cluster to the cells outside the cluster, as Implemented in the scanpy function tl.Rank_genes_groups. The same was done to identify Differentially expressed genes Between pairs of clusters.

## Ethical statement

Fish were kept according to FELASA guidelines (*Aleström et al., 2020*; *Brand et al., 2002*). All animal experiments were conducted according to the guidelines and under supervision of the Regierungspräsidium Dresden (permit: TVV 44/2017). All efforts were made to minimize animal suffering and the number of animals used.

## Acknowledgements

We thank Dr. Judith Konantz, M Fischer, S Kunadt and D Mögel for excellent zebrafish care. We thank Katja Bernhardt for the excellent technical assistance in flow cytometry and cell sorting. We would like to thank also Susanne Reinhardt and Dr. Andreas Petzold from the DRESDEN-concept Genome Center for support with initial planning and bioinformatic processing of the scRNAseq experiment, respectively. We thank Prof. Marius Ader, Prof. Mike O Karl, Dr. Volker Kroehne, Juliane Hammer and Elisa Nerli for critically reading the manuscript. In addition, we thank the members of the Brand laboratory for continued support and discussions as well as helpful comments on the manuscript. We thank also Dr. Christian Lange, Dr. Robert Münch and Dr. Steffen Rulands for excellent scientific discussion. This work was supported by grants to M.B. from the Deutsche Forschungsgemeinschaft (BR 1746/11-1), the European Union (European Research Council AdG Zf-BrainReg) and the German Excellence Initiative (Wissenschaftsrat and Deutsche Forschungsgemeinschaft) (EXC 168). Work was supported by the DRESDEN-concept Genome Center, the Light Microscopy Facility and Flow Cytometry Facility, core facilities of the CMCB at the Technische Universität Dresden. The funders had no role in study design, data collection and analysis, decision to publish, or preparation of the manuscript.

## Additional information

### Funding

| Funder | Grant reference number | Author |
| --- | --- | --- |
| Deutsche Forschungsgemeinschaft | BR 1746/11-1 | Michael Brand |
| HORIZON EUROPE European Research Council | ERC AdG Zf-BrainReg | Michael Brand |
| Deutsche Forschungsgemeinschaft | EXC 168 | Michael Brand |

The funders had no role in study design, data collection and interpretation, or the decision to submit the work for publication.

## Author contributions
Laura Celotto, Conceptualization, Data curation, Software, Formal analysis, Validation, Investigation, Methodology, Writing – original draft, Writing – review and editing; Fabian Rost, Conceptualization, Resources, Data curation, Software, Formal analysis, Validation, Investigation, Methodology, Writing – original draft, Writing – review and editing; Anja Machate, Juliane Bläsche, Validation, Investigation, Methodology, Writing – review and editing; Andreas Dahl, Resources, Investigation, Methodology, Writing – review and editing; Anke Weber, Conceptualization, Writing – review and editing; Stefan Hans, Conceptualization, Software, Investigation, Writing – original draft, Writing – review and editing; Michael Brand, Conceptualization, Resources, Software, Funding acquisition, Investigation, Writing – original draft, Project administration, Writing – review and editing

## Author ORCIDs
Fabian Rost ⓘ https://orcid.org/0000-0001-6466-2589
Andreas Dahl ⓘ http://orcid.org/0000-0002-2668-8371
Michael Brand ⓘ http://orcid.org/0000-0001-5711-6512

## Ethics
Fish were kept according to FELASA guidelines (Aleström et al., 2020; Brand et al., 2002). All animal experiments were conducted according to the guidelines and under supervision of the Regierungspräsidium Dresden (permit: TVV 44/2017). All efforts were made to minimize animal suffering and the number of animals used.

Reviewer #1 (Public Review): https://doi.org/10.7554/eLife.86507.3.sa1
Reviewer #2 (Public Review): https://doi.org/10.7554/eLife.86507.3.sa2
Author Response https://doi.org/10.7554/eLife.86507.3.sa3

# Additional files

## Supplementary files
• Supplementary file 1. Top 100 marker genes per cluster.

• Supplementary file 2. Top 100 differentially expressed genes between Non-reactive Müller glia (1) and Non-reactive Müller glia (2).

• Supplementary file 3. Top 100 differentially expressed genes between Reactive Müller glia (1) and Reactive Müller glia (2).

• Supplementary file 4. Top 100 differentially expressed genes between Reactive Müller glia (1) and Müller glia/Progenitors 1.

• Supplementary file 5. Top 100 differentially expressed genes between Reactive Müller glia (2) and Müller glia/Progenitors 1.

• Supplementary file 6. Top 100 differentially expressed genes between Müller glia/Progenitors 1 and Progenitors 2.

• Supplementary file 7. Sequence based reagents.

• MDAR checklist

## Data availability
The raw data can be accessed at GEO under the accession number GSE226373. All sequencing data that support the findings of this study have been deposited at https://singlecell.broadinstitute.org/single_cell/study/SCP1973. The source code for the analysis is available on GitHub (https://github.com/fbnrst/celotto_2023; copy archived at *Rost, 2023*).

The following datasets were generated:

| Author(s) | Year | Dataset title | Dataset URL | Database and Identifier |
|---|---|---|---|---|
| Celotto L, Rost F | 2023 | Study: scRNAseq unravels the transcriptional network underlying zebrafish retina regeneration | https://singlecell.broadinstitute.org/single_cell/study/SCP1973 | Broad Institute, SCP1973 |
| Celotte L, Rost F, Machate A, Bläsche J, Dahl A, Weber A, Hans S, Brand M | 2023 | Single cell RNA sequencing unravels the transcriptional network underlying zebrafish retina regeneration | https://www.ncbi.nlm.nih.gov/geo/query/acc.cgi?acc=GSE226373 | NCBI Gene Expression Omnibus, GSE226373 |

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
