## [Editor Report · eLife assessment]

Müller glial cells of the zebrafish retina can differentiate into all neural cell classes following injury, providing full regenerative capabilities of the zebrafish retina. This **valuable** study presents a description of transcriptional changes of Müller glia cells in the adult and regenerating retina using single-cell RNA sequencing. The overall evidence supporting the main claims of the authors is **solid**.

---

## [Referee Report · Reviewer #1 (Public Review)]

Muller glia function as retinal stem cells in the adult zebrafish retina. Following retinal injury, Muller glia are reprogrammed (reactive Muller glia), and then divide to produce a progenitor that amplifies and differentiates into retinal neurons. Previous scRNAseq analysis used total retinal RNA from uninjured and injured retinas isolated at time points when Muller glia are quiescent, being reprogrammed, and proliferating to reveal genes and gene regulatory networks underlying these events (Hoang et al., 2020). The manuscript by Celotto et al., used double transgenic zebrafish that allow them to purify by FACS quiescent and reactive Muller glia, Muller glia-derived progenitors, and their differentiating progeny at different times post retinal damage. RNA from these cell populations was used in scRNAseq studies to identify the transcriptomes associated with these cell populations. Importantly, they report two quiescent and two reactive Muller glia populations. These results raise the interesting possibility that Muller glia are a heterogeneous population whose members may exhibit different regenerative responses to retinal injury. However, without further experimentation, the validity and significance of this result remain unclear. In addition to putative Muller cell heterogeneity, Celotto et al., identified multiple progenitor classes, some of which are specified to regenerate specific retinal neuron types. Because of its focus on Muller glia and Muller glia-derived progenitors at mid to late stages of retina regeneration, this new scRNAseq data will be a useful resource to the research community for further interrogation of gene expression changes underlying retina regeneration.

---

## [Referee Report · Reviewer #2 (Public Review)]

In this publication, the authors provide a comprehensive trajectory of transcriptional changes in Müller glia cells (MG) in the regenerating retina of zebrafish. These resident glia cells of the retina can differentiate into all neural cell classes following injury, providing full regenerative capabilities of the zebrafish retina. The authors achieved this by using single-cell RNA sequencing of Müller glia, progenitors, and regenerated progeny, comparing uninjured and light-lesioned retinae.

The isolation strategy involves using two transgenic strains, one labelling dividing cells and their immediate progeny, and the other Müller glia cells. This allowed them to separate injury-induced proliferating and non-reactive Müller glia cells. Subsequent single-cell transcriptomics showed that MG could be non-reactive under both uninjured and lesioned conditions and reactive MG gives rise to a cell population that both replenishes the pool of MG and replenishes neurogenic retinal precursor cells. These precursor cells produce regenerated neurons in a developmental time series with ganglion cells being born first and bipolar cells being born last. Interestingly hybrid populations have been detected that co-share characteristics of photoreceptor precursors and reactive glia.

This is the first study of its kind following the dynamic changes of transcriptional changes during retinal regeneration, providing a rich data source of genes involved in regeneration. Their finding of transcriptionally separable MG populations is intriguing.

This study focuses on the light-lesioned retina and leaves open the question if the observed transcriptional trajectories of regenerating neurons are generalizable to other lesion models (e.g. chemical or mutational lesions) or are specific to the light-damaged retina.

---

## [Author Response]

The following is the authors’ response to the original reviews.

**Reviewer #1 (Recommendations for The Authors)**
MAJOR CONCERNS1. Not addressed, but perhaps relevant, is that most of the postembryonic fish growth results from stem cells located in the ciliary marginal zone that make new neurons and Muller glia throughout the fish's life. Thus, Muller cell heterogeneity may result from the central to the peripheral gradient of Muller glial cell maturation.1a. Müller glial cell heterogeneity needs to be confirmed using, for example, in situ hybridization studies with gene-specific probes identified in the scRNAseq that distinguish these 2 populations. An additional approach could be the use of transgenic lines harboring tagged endogenous or transgene that reflects the promoter activity of the Muller glia subtypespecific gene.

We thank the reviewer for the insightful comments and agree on the importance to substantiate the Müller glia heterogeneity in our manuscript. Our study is not the only study that provides evidence for Müller glia heterogeneity. In particular, we would like to refer to a recent publication (Krylov et al., 2023). Using single cell RNA sequencing, Krylov et al. detect Müller glia heterogeneity in the uninjured retina, as well as upon selective, genetic ablation of distinct subtypes of photoreceptors (e.g. long and short wavelength sensitive cones, as well as rods). They observe six distinct clusters of quiescent Müller glia that show differential spatial distribution along the dorsal/ventral retinal axis. For instance, they report a ventral quiescent Müller glia population that shares some marker genes (aldh1a3, rdh10a, smoc1) with our nonreactive Müller glia 2 (cluster 2, supplementary files 1 and 2). Moreover, the authors report that Müller glia located at different positions along the dorsal/ventral axis exhibit distinct patterns of pcna upregulation as well as subsequent re-activation upon photoreceptor ablation. We have added the supportive information from Krylov et al. in the discussion section (lines: 781-789) of our manuscript.

1. Most interesting, but also least substantiated, is the authors' report of 2 different quiescent Muller glial cell populations in the uninjured retina and 2 different reactive Muller cell populations in the injured retina. If these populations exist independently of each other, it would be important to investigate if they differentially impacted retina regeneration.2a. CRISPR knockdown in F0 of factors thought to be involved in specific Müller glia-derived progenitor trajectories would be important to lend some functional significance to the data.

We fully agree with the reviewer that addition of functional data would enrich the manuscript with valuable information. However, we don´t believe that the suggested CRISPR knockdown of selected genes in F0 animals (also known as crispants) represents a suitable approach. Crispants have been used successfully to investigate genetic contributions in embryonic-tolarval stages (the first few days) of zebrafish development, as injection of multiple gRNAs targeting the same gene is sufficient to achieve a bi-allelic knockout of the gene of up to 90% (Kroll et al., 2021). However, unless both alleles of the target gene(s) is/are mutated already early on with nearly 100%, it is unlikely that the gRNA inactivation would work equally well during subsequent development into adult stages (several months later, and after exponential growth and volume increase of the animal). Even if biallelic inactivation in the crispants does work early on, it remains unclear whether and how crispants survive to adulthood, which will be necessary in order to address gene function in the context of retina regeneration. Moreover, since we observe that the genetic events during adult retina regeneration are highly similar to the events during retina development, we would rather expect the crispants already display developmental phenotypes, which would further hamper the study of potential regenerationspecific phenotypes in adult animals. We are convinced that only ‘clean’ conditional gene inactivation studies will be suitable to address the impact of Müller glia and derived progenitor trajectories on retina regeneration. In this respect, we have recently developed the new conditional Cre-Controlled CRISPR mutagenesis system (Hans et al., Nature Comm 2021). We are currently establishing stable lines to enable controlled and specific gene inactivation, but have only obtained preliminary results so far; the final analysis will take much more time and is, therefore, beyond the scope of this work.

1. The discussion should be modified to relate the data here presented with those described in Hoang et al., 2020.

We followed the suggestions of the reviewer and compared our single cell RNA sequencing dataset to that described in Hoang et al., 2020. As one might expect, the comparison between the two datasets showed similarities but also significant differences due to the different experimental set-ups. We show the results of this comparison in additional main (new Figure 9) and supplementary figures (new Figure 9-figure supplement 1). In order to compare our newly obtained scRNAseq dataset of MG and MG-lineage-derived cells of the regenerating zebrafish retina to the previously published dataset of light-lesioned retina (Hoang et al., 2020), we employed the ingestion method (Scanpy, https://scanpy-tutorials.readthedocs.io/en/latest/ integrating-data-using-ingest.html) and mapped the clusters identified by Hoang and colleagues to our clusters (new Figure 9). While we applied a short-term lineage tracing strategy and only sequenced the enriched population of FAC-sorted MG and MG-derived cells of the regenerating zebrafish retina, Hoang and colleagues sequenced all retinal cells in the light-lesioned retina. Consequently, comparison between the two datasets uncovered similarities, but also significant differences, due to the different experimental set-ups (Figure 9A). Consistently, the cluster annotated as resting MG in Hoang et al. mapped to clusters annotated as non-reactive MG 1 and 2 in our dataset (new Figure 9B). The cluster annotated as activated MG in Hoang et al. mapped to clusters annotated as reactive MG 1 and 2, as well as to the cluster with hybrid identity of MG/progenitors in our dataset. Interestingly, some cells annotated as activated MG in Hoang et al. mapped also to neurogenic progenitor 2 and 3 clusters in our dataset (Figure 9B). The cluster annotated as progenitors in Hoang et al. mapped to the progenitor area in our dataset, which included neurogenic progenitors 2, 3 as well as photoreceptor and horizontal cell precursors (new Figure 9B). Finally, retinal ganglion cells, cones, GABAergic amacrine cells and bipolar cells annotated in Hoang et al. perfectly mapped to retinal ganglion cells, cone, amacrine and bipolar cells in our dataset (new Figure 9B). While we did not detect a mature horizontal cell cluster, Hoang and colleagues annotated a horizontal cell cluster, which cells mapped to reactive MG 2, MG/progenitors 1 and part of progenitors 3 in our dataset (new Figure 9B). Moreover, Hoang and colleagues annotated rod photoreceptors that mapped to progenitors 3, photoreceptor precursors, red and blue cones, horizontal cell precursors and bipolar cells in our dataset (new Figure 9B). Finally, due to the different cell isolation protocol, Hoang and colleagues annotated additional cell clusters that did not map to any cluster in our more selective dataset, and included oligodendrocytes, pericytes, retinal pigmented epithelial cells as well as vascular/endothelial cells (new Figure 9B). Next, we selected representative marker genes per cluster from our scRNAseq dataset and checked their expression in the dataset by Hoang and colleagues (Figure 9-figure supplement 1). The dot plot showing the expression of selected gene candidates per cluster further corroborated the large overlap between clusters annotated in the present study with those annotated in the study by Hoang and colleagues. These novel comparisons to the data of Hoang et al. are now included in the resubmitted version, and are described and discussed in an additional paragraph in the results (lines: 482-517) as well as discussion (lines: 766-807) sections.

MINOR CONCERNS1. Fig 1C is difficult to interpret. I am also confused by the color coding which is not presented in the figure legend - why 3 shades of red and two of blue? Please define each (for example, what's the difference between red, purple, and light red in the 6dpl panel?). What are the white areas outlined by blue and red circles/cells (looks like a topography plot)? It appears that there is a fairly large amount of pcna:EGFP expression in the uninjured retina - what are these cells?

We have replaced Figure 1C with a better one and rephrased/extended the explanation of the figure in the results (lines: 192-195). Figure 1C depicts contour plots, which represent the relative frequency of data. Each contour line encloses an equal percentage of events (that is, cells), and contour lines that are closely packed indicate a high concentration of events. In flow cytometry, contour plots are used to represent highly frequent events, as this kind of plots are independent on sample size.

Concerning the observed pcna:EGFP expressing cells in the uninjured retina, we interpret them as proliferating cells coming from the ciliary marginal zone and from Müller glia of the central retina, which represent progenitors and Müller glia that have re-entered the cell cycle to generate rod progenitors, respectively. Consistent with that, we observe pcna:EGFPpositive cells in the ciliary marginal zone as well as central retina using immunofluorescence, as shown in Figure 1-figure supplement 1.

1. Results, lines 186-188 are not presented clearly: EGFP+ cells may persist for some time after they leave the cell cycle, so stating EGFP+ cells are proliferating may not be correct. How long does PCNA promoter activity and EGFP expression remain after Muller cells exit the cell cycle?mCherry+/EGFP- cells may be non-reactive Muller glia or reactive Muller glia that have not entered the cell cycle. It seems likely that Muller glia start reprogramming before undergoing cell division.

We agree with the reviewer that EGFP persists for some time after the cells have left the cell cycle, which we actually describe and use to benefit in our study. We do not know for how long exactly the pcna promoter is active within the cell cycle, but EGFP is known to have a half-life of approximately 24 hours (Li et al., 1998). Even though we cannot make a statement about EGFP persistence in Müller glia, we note that previous reports (Lahne et al., 2015; Nagashima et al., 2013; Nelson et al., 2013; Thummel et al., 2008) and our study (Figure 3-figure supplement 2) show PCNA at the protein level in Müller glia cells between 24 and 48 hpl, including our sampled 44 hpl time point (lines: 69-73). We also agree with the reviewer that Müller glia will become reactive to the injury most likely prior (lines: 67-69) to activation of the pcna promoter, meaning that Müller glia are EGFP-negative at this time point due to the immature status of EGFP after translation. However, we are confident that our data also comprises this cell state (early phase of Müller glia activation) because we sampled proliferating (EGFP- and mCherry-double positive cells) as well as non-proliferating Müller glia (mCherry-only positive cells) at all time points (lines: 213-215 and Figure 1C). We interpret that the early phase of Müller glia activation corresponds to Müller glia transitioning from a nonreactive to a reactive state. With respect to our UMAP, we map this cell state in cluster 1 localizing to the top left part of the cluster, abutting cluster 3, the reactive Müller glia 1 (Figure 2B).

1. I am concerned by the observation that microglia were identified by scRNAseq as a contaminating cell population. Since FACS was based on gfap:mCherry expression, why did microglia end up in the mix? Also, what are the ‘...low-quality cells expressing many ribosomal transcripts...’ and why, if they are low-quality cells, did they pass the sequencing quality control as stated on lines 208-209?

The reviewer is right that microglia should actually not end up in the sample when using the gfap:mCherry line. However, microglia always displayed a certain level of autofluorescence in our experimental set-up (possibly because they may have ingested some cell debris), which may have contributed to their presence in the FACS samples. In contrast to the reviewer, we were not concerned about this ‘contamination’, because the microglia could be easily identified and sorted out using bioinformatics. This is supported by the presented supplementary figure in which microglia separate from the core of clusters containing Müller glia and Müller gliaderived cells in the UMAP of the full dataset (Figure 2-figure supplement 1).

We also acknowledge that ‘low quality cells’ is not an appropriate term for cells in the cluster expressing ribosomal mRNAs at high levels, as ribosomal enrichment actually does not give any information concerning their quality. We referred to them as ‘low quality’ because the enrichment in ribosomal transcripts masks their identity considerably. To correct this, we now renamed cells in this cluster descriptively as ‘ribosomal gene-enriched’ cells (Figure 2-figure supplement 1, line: 226).

1. Fig. 2: please list in the text or fig legend the specific genes used to identify each cell cycle state. Why is cluster 3 considered a reactive Muller population when expressing S phase markers and PCNA? These features seem to distinguish cluster 3 from 4 and may suggest cluster 3 is a progenitor population. Further explanation is necessary to understand the assignments.

Information about the specific genes used to identify each cell cycle state is provided in the paragraph “Bioinformatic analysis” (lines: 925-934) in the Materials and Methods section. We considered listing all the markers in either the results or the figure legends as well, but decided against it, as it impairs readability in our opinion. Nevertheless, we have now highlighted also in the results (line: 261) that the list of cell cycle markers is available in the Materials and Methods section.

We understand the reviewer´s point that cluster 3 represents progenitors and not Müller glia, when PCNA expression is considered as a sole marker of progenitors or of Müller glia reprogrammed to a progenitor state (Hoang et al., 2020). However, we disagree with this view for three reasons. First, although PCNA is used as a marker of Müller glia reprogrammed to a progenitor state and of progenitors in Hoang et al., 2020, it should be noted that PCNA-positive, Müller glia cells are present in the central retina already in uninjured conditions, when regeneration-associated, Müller glia-derived progenitors are not detectable. Second, cluster 3 is evident only at 44 hpl, a time point at which Müller glia cells are about to divide or have undergone their first and only cell division. In this regard, we would like to refer to the discussion about Müller glia and Müller glia-derived progenitors as distinct populations in Lenkowski and Raymond, 2014. Third, we have performed in situ hybridization for starmaker (stm), a marker gene highly specific for cells in cluster 3 (supplementary files 1 and 3), combined with immunohistochemistry for GFAP and PCNA. The results of the staining are depicted in a new Figure 3-figure supplement 2. In strong agreement with our sequencing results, we observe stm expression only at 44 hpl, whereas no signal is detected in the uninjured as well as 4 and 6 dpl retina (Figure 3- figure supplement 2). Virtually all stm-positive cells at 44 hpl are also PCNA- and GFAP-double positive cells displaying a clear Müller glia morphology (Figure 3- figure supplement 2). Hence, we interpret cells in cluster 3 as reactive Müller glia, indicating that pcna can be used as a marker of progenitors, but not exclusively of progenitors, prevalently at later stages. At 44 hpl, Müller glia express pcna in order to undergo asymmetric cell division giving rise to the renewed Müller glia and the multipotent progenitor that will continue dividing.

1. I am confused by the crlf1a scRNAseq data indicating it is associated with proliferating PCNA+ reactive Muller glia Cluster 3 and PCNA- reactive Muller glia Cluster4 at 44 hpl (Fig. 3), yet in Fig. 4 crlf1a in situ signal is exclusively associated with proliferating Muller glia at 44 hpl. Why don't we observe the crlf1a+/PCNA- cell population?

We highlight that crlf1a expression is actually detected also at 4 dpl (Fig. 3). We also note that immunofluorescence in Fig 3. shows crlf1a mRNA and PCNA protein, whereas single cell RNA sequencing detects crlf1a and pcna transcripts. In this context, it is possible that crlf1a-, PCNAdouble positive cells detected at 4 dpl are still positive for the PCNA protein, but no longer express the pcna transcript. Double in situ hybridization for pcna and crlf1a would be needed to fully address whether crlf1a-positive cells are still pcna-positive at 4 dpl. It is also possible that crlf1a-, GFAP-double positive, PCNA-negative Müller glia are fewer and only masked in the crowd of crlf1a-, PCNA-double positive, GFAP-negative progenitors at 4 dpl (Raymond et al., 2006). We amended the discussion section with this information (lines: 634-654).

1. scRNAseq cluster 3 is a proliferating population that is assigned "reactive Muller glia", whereas cluster 5 is assigned Muller glia/progenitor and in the Discussion referred to as MG about to go or already underwent asymmetric division to generate a progenitor (lines 568-571). I don't understand why cluster 3 is not referred to as the one harboring reactive MG/progenitors that underwent or are undergoing asymmetric cell division - The timing is right, as are the markers.

We would like to refer the reviewer to the discussion in point 4, including the changes we introduced in the Materials and Methods (Lines 925-934). As mentioned above, we do not agree that PCNA alone represents an exclusive marker of progenitors, but is rather a marker of cells undergoing proliferation. Moreover, we note that Müller glia first and only division occurs between 31 and 48 hpl. Finally, as mentioned above, expression of stm is a unique marker for cluster 3, which is only evident at 44 hpl, but not of cluster 5, which is evident at 4 dpl.

It seems cluster 5 might better fit the amplifying progenitor stage where some MG markers are retained but diluted by cell division. Please clarify the reasoning behind the labeling of this cluster. It is not clear why this cluster has to contain self-renewed Muller glia - why wouldn't these Muller cells partition to quiescent MG clusters 1 and 2 or reactive Muller glia in clusters 3 and 4?

We partially agree with the reviewer that cluster 5 might better fit the amplifying progenitor state, and this is why we indicate this cluster as a “crossroad in the trajectory” in the discussion (lines: 613-631). However, we cannot entirely exclude that cells in cluster 5 contain selfrenewed Müller glia (differential gene expression analysis highlights glial markers too, see Figure 3A, supplementary file 6). Cells that we interpret as self-renewing Müller glia do not partition back to quiescent Müller glia (cluster 1 and 2) because they are on the way to be quiescent Müller glia again, yet they did not reach that point, maybe due to sampling reasons. Unfortunately, our short-term lineage tracing strategy ceases at 6 dpl. We also speculate in the discussion (lines: 679-682) that if we had sampled at later time points (e.g. at 14 dpl), we might have been able to detect the density of the cells in the glial area moving back to clusters 1 or 2 (cell density plots, Figure 2B).

I also have trouble understanding cluster 4's assignment. The Discussion states it represents cells at the crossroad of glial and neurogenic trajectory containing self-renewed Muller glia as well as first-born MG-derived progenitors. However, it is populated by cells after 44 hpl (Fig. 2B) which is when reactive Muller glia are detected and lacks proliferative markers.

We think that there is a misunderstanding here. We never refer to cluster 4 as a crossroad in the glial and neurogenic trajectory. We state that cluster 5 is actually the crossroad between the two trajectories (line 629). We further propose that self-renewed MG close the cycle via late reactive MG (cluster 4) and return into non-reactive Müller glia (clusters 1 and 2, red, dashed line in Figure 10) (now described in lines 631-633). The cell density plots support the direction of the cycle closing towards non-reactive Müller glia, in particular at 4 and 6 dpl (Figure 2B).

Might cluster 4 represent a population of reactive MG remaining at 4 dpl that never entered the cell cycle and therefore would be devoid of Muller glia-derived progenitors?

As stated in the manuscript, we actually think that marker expression as well as the cell density plots support our assignment of cluster 4 to represent self-renewed Müller glia closing the cycle to non-reactive Müller glia. Our assignment also fits well with the expected events following asymmetric cell division. However, as we cannot rule out the reviewer´s entire idea, we included the suggestion in the updated discussion (lines 651-654).

1. Results, lines 163-164; Please provide a reference for "..... consistent with the previously described....."

We thank the reviewer for this observation and we added the appropriate references (Fimbel et al., 2007; Lenkowski and Raymond, 2014; Thummel et al., 2008) in the updated version of the manuscript (lines: 171-172).

**Reviewer #2 (Recommendations For The Authors):**
Overall, this very thorough study provides interesting and unexpected results. The published data set will be useful for many subsequent studies. I have only a few remarks that the authors may consider discussing. Their cluster analysis revealed most of the expected cell clusters with some interesting surprises. One relates to photoreceptors where the authors describe well-separated clusters for red and green cones, while rods, UV and blue cones do not form clusters. For rods, this is discussed, but I miss a brief discussion on the "missing" cone subtypes.

We thank the reviewer for the insightful comments. It is correct that we indeed detect only red and blue cones, as indicated by their expression of red-sensitive opsin gene (opn1lw2) and the blue-sensitive opsin gene (opn1sw2), respectively. It is possible that missing cone subtypes are born later than 6 dpl. As the reviewer suggested, we amended the discussion and added information about the missing cone subtypes (lines: 724-726).

I am also intrigued by the two, quite separated amacrine cell clusters, while bipolar cells cluster in one cluster, without separation in (say) ON and OFF bipolar cells. This may also merit a discussion. What are their ideas on the small and quite separated amacrine cell cluster (cluster 14).

Bipolar cells in cluster 15 are very sparse in our dataset, with only 40 cells in total. Hence, the sample size might be too small to be separated into ON and OFF subtypes. Alternatively, cells might be still immature, as we use 6 dpl as our latest sampled time point. Concerning cells in cluster 14, we think they are starburst amacrine cells, as indicated by their simultaneous expression of gad1b and chata (Figure 8-figure supplement 2B), which is a characteristic feature of starburst amacrine cells in mouse (O´Malley et al., 1992). We added this observation in the discussion (lines: 706-712).